# Reduced secretion of neuronal growth regulator 1 contributes to impaired adipose-neuronal crosstalk in obesity

Elisa Duregotti [1], Christina M. Reumiller [1], Ursula Mayr[1], Maria Hasman [1], Lukas E. Schmidt [1], Sean A. Burnap [1], Konstantinos Theofilatos [1], Javier Barallobre-Barreiro[1], Arne Beran[2], Maria Grandoch[2], Alessandro Viviano[3,4], Marjan Jahangiri[3] & Manuel Mayr [1] ✉

While the endocrine function of white adipose tissue has been extensively explored, comparatively little is known about the secretory activity of less-investigated fat depots. Here, we use proteomics to compare the secretory profiles of male murine perivascular depots with those of canonical white and brown fat. Perivascular secretomes show enrichment for neuronal cell-adhesion molecules, reflecting a higher content of intra-parenchymal sympathetic projections compared to other adipose depots. The sympathetic innervation is reduced in the perivascular fat of obese (*ob/ob*) male mice, as well as in the epicardial fat of patients with obesity. Degeneration of sympathetic neurites is observed in presence of conditioned media of fat explants from *ob/ob* mice, that show reduced secretion of neuronal growth regulator 1. Supplementation of neuronal growth regulator 1 reverses this neurodegenerative effect, unveiling a neurotrophic role for this protein previously identified as a locus associated with human obesity. As sympathetic stimulation triggers energy-consuming processes in adipose tissue, an impaired adipose-neuronal crosstalk is likely to contribute to the disrupted metabolic homeostasis characterising obesity.

Adipose tissue (AT) is a heterogenous connective tissue composed of mature adipocytes, non-adipocytes, capillaries and nerves. In mammals, AT is traditionally classified into two main types: white AT (WAT) and brown AT (BAT). While the primary task of WAT is the storage of triglycerides[1], BAT represents the main site of non-shivering thermogenesis, where lipids are burnt to generate heat in a process mediated by the BAT-specific uncoupling protein 1 (UCP1)[2]. The sympathetic nervous system (SNS) is increasingly being recognised as a master regulator of metabolic processes in AT, including lipolysis[3], thermogenesis[2], and browning—i.e., the recruitment of brown adipocytes within WATs[4]. The activation of these energy-dissipating processes has been associated with protective effects against obesity and related metabolic disorders[5,6]. Thus, efforts are currently being directed towards the characterisation of browning mechanisms, which could reveal novel therapeutic targets[7].

Besides its role as an energy storage site, AT is a complex endocrine organ secreting a variety of bioactive peptides named adipokines[8]. Protein secretion differs between WAT and BAT in line with their different metabolic functions[9,10]. Some of the most important adipokines include hormones (i.e., leptin, adiponectin, resistin),

[1]King's College London British Heart Foundation Centre, School of Cardiovascular Medicine and Sciences, London, UK. [2]Institute of Translational Pharmacology, University Hospital Düsseldorf, Heinrich-Heine-University Düsseldorf, Düsseldorf, Germany. [3]Department of Cardiothoracic Surgery, St. George's Hospital, University of London, London, UK. [4]Present address: Department of Cardiothoracic Surgery, Hammersmith Hospital, Imperial College London, London, UK. ✉e-mail: manuel.mayr@kcl.ac.uk

cytokines (i.e., IL-6, TNFα, CCL2) and growth factors (i.e., IGF, VEGF, PDGF) that exhibit endocrine, autocrine and paracrine functions, thus making AT a major regulator of energy homeostasis[10]. Dysregulations in the adipokines profiles have been observed in individuals with obesity, linking obesity to associated metabolic and cardiovascular disorders[10,11].

In addition to canonical AT depots, perivascular adipose tissue (PVAT) surrounds most blood vessels and has been implicated in the crosstalk with the underlying vascular wall[12]. Through the release of soluble factors, PVAT is involved in the paracrine modulation of vascular tone[13,14], blood pressure[15] and intravascular temperature[16]. Akin to other fat depots, perivascular adipocytes are affected by obesity, but to date only a few studies have been conducted to examine the PVAT in this context[17–20]. Moreover, a comprehensive characterisation of the PVAT secretory profile has yet to be described and could provide important insights regarding changes in the perivascular environment in obesity.

In this study, we employed proteomics to compare the secretory profiles of distinct perivascular depots with those of canonical WATs and BAT. Proteins related to axon guidance were enriched in conditioned media from PVAT, reflecting a denser sympathetic innervation compared to other depots. The phenotype and secretory profile of PVAT were substantially altered in *ob/ob* mice, which displayed an impaired innervation of PVAT, subcutaneous WAT and BAT. This observation was confirmed in the epicardial fat of humans, with obesity leading to a significant reduction of axonal markers. Proteomics revealed attenuated secretion of neuronal growth regulator 1 (Negr1) by AT from obese mice. Experiments on murine primary sympathetic neurons and in vivo administration of recombinant Negr1 confirmed that this adipokine stimulates axonal growth and protects against neurodegeneration. Thus, Negr1 is a key molecular factor in the disrupted adipose-sympathetic crosstalk in obesity.

## Results

### Distinct AT depots display unique secretory profiles

Three canonical and 3 perivascular fat depots from wild type (wt) C57BL/6 J mice were used in the present study: the interscapular BAT, subcutaneous (SC) and visceral (VI) WAT and the PVAT surrounding the arch (AR), thoracic (TH) and abdominal (AB) aorta (Fig. 1a). Consistent with previous evidence[17,21], the AB PVAT resembled the phenotype of WATs, being composed of adipocytes filled with a single large lipid droplet (Fig. 1b, upper panel). In contrast, the AR and TH PVAT consists of smaller multilocular adipocytes, similar to those observed in the interscapular BAT (Fig. 1b, upper panel). These phenotypical differences were confirmed by immunohistochemistry: staining for perilipin highlighted the presence of numerous small BAT-like lipid droplets in AR and TH PVAT, that also express the brown-specific marker UCP1[2] (Fig. 1b, lower panel and Fig. 1c). On the other hand, UCP1 was undetectable in WATs and AB PVAT (Fig. 1b, lower panel and Fig. 1c), that shared a similar staining pattern for perilipin, outlining large lipid droplets occupying most of the cytosol of adipocytes (Fig. 1b, lower panel).

For proteomics, AT explants were incubated in serum-free culture medium for 24 h (Fig. 1a). Protein secretion differed across distinct AT depots as evidenced by Coomassie staining of their supernatants (Supplementary Fig. 1a). Immunoblotting confirmed that secreted adipokines–such as adiponectin–were much more abundant in conditioned media compared to intracellular proteins that were enriched in AT lysates (Fig. 1d). Subsequently, 6 biological replicates (N = 6 mice) for each depot were analysed by liquid chromatography tandem mass-spectrometry (LC-MS/MS). The proteomics analysis returned a total of 2407 non-redundant proteins, of which 526 were predicted to be secreted by the SignalP prediction software[22] or the Matrisome database[23] (Supplementary Data 1). Well-established adipokines such as adiponectin (Adipoq), adipsin (Cfd), resistin (Retn), chemerin

(Rarres2), retinol-binding protein 4 (Rbp4), lipocalin-2 (Lcn2), visfatin (Nampt) and plasminogen activator inhibitor-1 (Serpine1)[11] were identified (Supplementary Data 1), with some of them quantitatively validated by immunoblotting (Supplementary Fig. 1b). Unsupervised hierarchical clustering of 363 consistently detected secreted proteins revealed that each fat depot is endowed with a characteristic secretory profile (Fig. 1e), mainly determined by its white or brown phenotype. Accordingly, the AB PVAT clustered together with SC and VI WATs, whereas TH and AR PVAT were more similar to BAT (Fig. 1e, f). However, proteins specifically enriched in the conditioned media of each depot were found to relate to distinct biological processes (Supplementary Fig. 1d), highlighting potential differences in terms of metabolic functions and cellular composition. Proteins enriched in WATs secretomes were mainly related to the immune-inflammatory response (Supplementary Fig. 1d), while as expected[9], BAT exhibited a lower secretory capacity (Fig. 1e). Among the few proteins with higher secretion by BAT were Visfatin (Nampt), S100b and Kallikrein-1 (Klk1) (Supplementary Fig. 1c), that have previously been related to BAT-specific regulatory processes[24–26].

### Perivascular secretory signature reveals a dense sympathetic innervation of PVAT depots

To further explore the secreted proteins that distinguish PVAT (Fig. 1e, white frame), we compared the PVAT secretory profiles (group 1, including AR, TH and AB PVAT samples, N = 18) with all the non-perivascular depots (group 2, including VI and SC WAT and BAT, N = 18) (Fig. 2a and Supplementary Data 1). Proteins with lower secretion by PVAT were involved in lipid transport (light-blue spots in Fig. 2a: i.e., apolipoproteins–Apoa1, Apoa4, Apod, retinol-binding protein 4–Rbp4 and glycosylphosphatidyl-inositol-anchored high-density lipoprotein-binding protein 1–Gpihbp1), blood coagulation and haemostasis (i.e., Plg) and immune-related processes (dark blue spots in Fig. 2a: i.e., several members of the serpin family) (Fig. 2a, b). This is in line with the main metabolic and endocrine regulatory functions of WAT (see also Supplementary Fig. 1d). On the other hand, proteins involved in cell-adhesion, cell differentiation and, unexpectedly, nervous system development and axon guidance were more abundant in the conditioned media of PVAT (Fig. 2a–red spots–and 2b). In particular, neural cell-adhesion molecule 1 (Ncam1), neural cell-adhesion molecule L1 (L1cam) and neural cell-adhesion molecule L1-like protein (Chl1) were found to be the most significantly upregulated proteins in the PVAT secretomes (Fig. 2a, e). Immunostaining on PVAT sections revealed that these proteins are not expressed by adipocytes (perilipin-positive), but rather display a fluorescence pattern compatible with that of intra-parenchymal neuronal arborisations[27] (Fig. 2c). Their co-localisation with the sympathetic marker tyrosine hydroxylase confirmed that these cell-adhesion molecules are expressed by sympathetic neurites embedded within the perivascular fat (Fig. 2d). Validation by immunoblotting (Fig. 2e, f) showed that only the smallest, soluble fragments of Ncam1, Chl1 and L1cam[28,29] were detected and increased in the conditioned media of PVAT samples. This indicates that their presence in secretomes is not due to membrane ruptures or axonal necrosis.

Considering the neuronal origin of these cell-adhesion molecules, we hypothesised that their higher abundance in the PVAT secretomes may reflect a greater parenchymal innervation of these depots. Since the SNS is mainly responsible for fat innervation[4], we probed the lysates and sections of all analysed AT depots for tyrosine hydroxylase. In line with previous reports[4,25,27,30], the interscapular BAT showed a higher content of sympathetic neurites with respect to canonical WATs (Fig. 2g, h), which have traditionally been described as poorly innervated. However, SC fat contains more sympathetic fibres than VI (Fig. 2g, h), as the SNS is involved in the recruitment of beige adipocytes embedded within the SC fat pad[27]. Surprisingly, tyrosine hydroxylase was even more abundant in PVAT than in BAT (Fig. 2g),

 

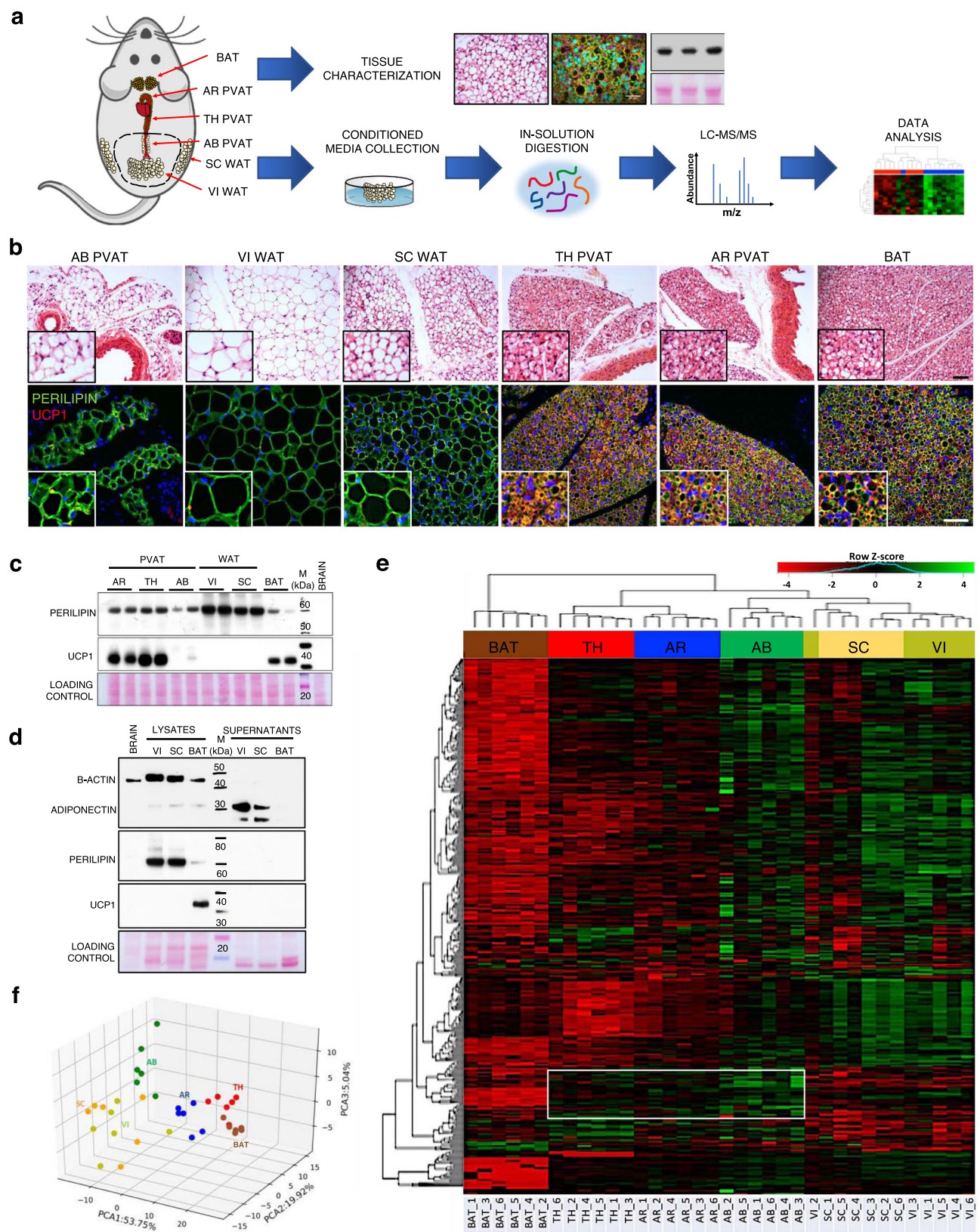

suggesting that a higher number of neuronal projections is a defining feature of PVAT depots. This was confirmed by immunostainings (Fig. 2h). PVAT from AR and TH regions displayed a denser distribution pattern of sympathetic arborisations with respect to all other AT samples. An exception, however, was the AB PVAT: the greater abundance of tyrosine hydroxylase detected in protein extracts was indeed

due to the presence of thick nerve fibres running alongside the aorta. The intra-adipose density of neurites in AB PVAT was very low and comparable to that observed in VI WAT (Fig. 2h). The intensity of β3-adrenoreceptor staining on the plasma membrane of adipocytes strongly correlates with the extent of innervation (Fig. 2h), consistent with a crosstalk between sympathetic inputs and adipocytes[2].

**Fig. 1 | Phenotypical characterisation and secretory profiling of different AT depots. a** Experimental workflow showing anatomical localisation of AT depots and subsequent procedures performed on AT explants. **b** Upper panel: H&E stainings of tissue sections. Lower panel: immunofluorescence images of AT sections stained for perilipin and UCP1. Nuclei are counterstained with DAPI. Scale bars: 50 μm. Insets show higher magnifications of randomly selected areas. Representative pictures of 1 out of 2 replicates. **c** Western blot of representative AT lysates (2 out of 4 replicates) stained for perilipin and UCP1. A brain lysate was loaded as negative control for AT-specific proteins. **d** Western blot of representative AT lysates (1 out of 2 replicates) and corresponding supernatants stained for β-actin, adiponectin, perilipin and UCP1. **e, f** Unsupervised hierarchical clustering (heatmap, **e**) and principal component analysis (**f**) of the 363 consistently identified secreted proteins detected by MS in the conditioned media of different AT depots. Hierarchical clustering was performed using the complete agglomeration method, *Z*-score-scaled abundances are displayed. *N* = 6 biological replicates per group. AT adipose tissue, AR PVAT aortic arch perivascular AT, TH PVAT thoracic PVAT, AB PVAT abdominal PVAT, VI WAT visceral white AT, SC WAT subcutaneous white AT, BAT interscapular brown AT, UCP1 uncoupling protein 1, M molecular weight marker.

## Obesity severely affects the phenotype and secretory profile of perivascular depots

To investigate how obesity impacts on the PVAT and affects its secretory profile, we dissected fat explants from wt and leptin-deficient *ob/ob* mice, a well-established model of obesity[31] (Supplementary Fig. 2a). As shown in Fig. 3a, b, the obese perivascular depots undergo substantial phenotypical changes with respect to their lean counterparts. Similar to VI and SC WAT, the adipocytes in the AB PVAT become severely hypertrophic, due to an increased intracellular fat accumulation (Fig. 3b). The AR and TH PVAT of *ob/ob* mice also displays larger lipid droplets compared to control animals, closely resembling the phenotypic changes observed in interscapular BAT (Fig. 3b).

Obesity also induced considerable changes in the secretomes of PVAT (Fig. 3c, d), with several well-established adipokines—such as adipsin (Cfd), resistin (Retn), complement C1q and tumour necrosis factor-related protein 9 (C1qtnf9) and retinol-binding protein 4 (Rbp4)—being significantly dysregulated compared to wt mice (green spots in Fig. 3d, Supplementary Data 2). Upregulated proteins in the secretomes of PVAT explants from *ob/ob* mice were related to lipid metabolism and transport processes (blue spots in Fig. 3d, chart in Fig. 3e). Among proteins with reduced secretion by PVAT from *ob/ob* mice were many cell-adhesion molecules involved in axon guidance (red spots in Fig. 3d, chart in Fig. 3e), including Ncam1, Chl1 and L1cam (Fig. 3d and Supplementary Fig. 2b). Their reduced abundances were validated by immunoblotting (Fig. 3f) and reflected decreased sympathetic inputs in PVATs from *ob/ob* mice, as evidenced by the reduced tyrosine hydroxylase content of the corresponding pooled PVAT lysates from *ob/ob* animals compared to controls (Fig. 3g). However, the difference observed in AB PVAT is probably a dissection-related artefact. In *ob/ob* mice, the sympathetic bundles contained in the AB PVAT are further from the aortic wall compared to wt, as a consequence of AT remodelling (Supplementary Fig. 2c). As the AB PVAT is not well-isolated from the surrounding VI WAT, these sympathetic nerves are lost during dissection in the attempt to only isolate AT from the perivascular region.

## Obesity is associated with an impaired sympathetic innervation of murine and human AT depots

To further investigate how the perivascular sympathetic innervation is affected by obesity, we probed AR and TH PVAT lysates and sections for tyrosine hydroxylase. Tyrosine hydroxylase was significantly decreased in both TH and AR PVAT samples from *ob/ob* mice compared to their wt counterparts (Fig. 4a, b). At a histological level, this reduction was paralleled by an altered distribution pattern of the tyrosine hydroxylase fluorescent signal: in PVAT sections from *ob/ob* mice the staining was dotted rather than continuous (Fig. 4c, d and Supplementary Movies 1 and 2). This fragmentation of sympathetic neurites was quantified by measuring the average size of tyrosine hydroxylase-positive particles, and the structural integrity of axons was found to be significantly reduced in PVAT from *ob/ob* animals (Fig. 4e). A significant reduction of tyrosine hydroxylase was also detected in SC fat lysates from *ob/ob* mice, accompanied by lower levels of UCP1 (Supplementary Fig. 3a), as well as in lysates from AB PVAT (Supplementary Fig. 3b) and from BAT (Supplementary Fig. 3c). Notably, tyrosine hydroxylase was also decreased in the AR and TH PVAT and in the SC WAT of diet-induced obese (DIO) mice (Supplementary Fig. 3d), indicating that a loss of adipose-sympathetic innervation also occurs in another model of murine obesity.

Considering the essential roles of the sympathetic stimulation in regulating the metabolism of AT, we decided to explore whether a similar loss of sympathetic neurites as observed in obese mice might also occur in human patients with obesity. For this purpose, we obtained epicardial fat surrounding the coronary arteries (Fig. 4f) during coronary artery bypass or aortic valve replacement surgery (Supplementary Table 1). Consistent with our findings in mice, the tyrosine hydroxylase content was significantly lower in the epicardial fat of patients with obesity (BMI ≥ 30, Fig. 4g, h). There was a significant negative correlation between sympathetic innervation of epicardial fat and BMI (Fig. 4i), but no significant effect of age, sex and pre-op administration of β-blockers (Supplementary Fig. 3e, f, g, respectively). A similar negative correlation was observed between BMI and another axonal marker, β3-tubulin (Fig. 4g, j), confirming that obesity is also associated with a decreased sympathetic innervation of human epicardial adipose tissue.

## Loss of sympathetic innervation in *ob/ob* AT is due to local microenvironment alterations

Since the sympathetic neurites are tightly embedded within the adipose parenchyma, we hypothesised that their impairment could be due to an altered surrounding microenvironment in obesity. To test this hypothesis, primary sympathetic neurons from superior cervical ganglia (SCGs) of C57BL/6 J wt neonatal mice were seeded on microfluidic devices that allow the physical compartmentalisation of cell bodies and axons (Fig. 5a and Supplementary Fig. 4a), thus mimicking the in vivo anatomical organisation where the neuronal somata are located away from fat depots, in the sympathetic ganglia. Once the axons had grown successfully (see Supplementary Fig. 4a), the growth medium in the axonal chamber was replaced with the conditioned media of fat explants obtained from either wt or *ob/ob* mice (Fig. 5a). As shown in Fig. 5b, 24 h after incubating with secretomes from *ob/ob* samples the neurites displayed a significantly higher number of varicosities, a hallmark of early stage neurodegeneration[32]. These swellings along axons were detectable both at a plasma membrane (Fig. 5b, left panel with live brightfield images) and at a cytoskeletal level (β3-tubulin staining in Fig. 5b, right panel and in Supplementary Fig. 4b showing lower magnification pictures of axonal chambers), while neuronal cell bodies in the somatic chamber remained unaffected (Fig. 5b, left). The tyrosine hydroxylase content was unchanged in the heart of *ob/ob* animals (Fig. 5c), reinforcing the notion that the neurodegeneration observed in AT depots is the result of a local dysfunctional microenvironment, rather than a central-mediated process. Given the close proximity of the PVAT and the vascular wall, however, decreased sympathetic innervation was also observed in PVAT-denuded aortas from *ob/ob* mice (Fig. 5d). Thus, the detrimental impact of fat secretome on neuronal processes extends beyond the PVAT to the underlying vessels.

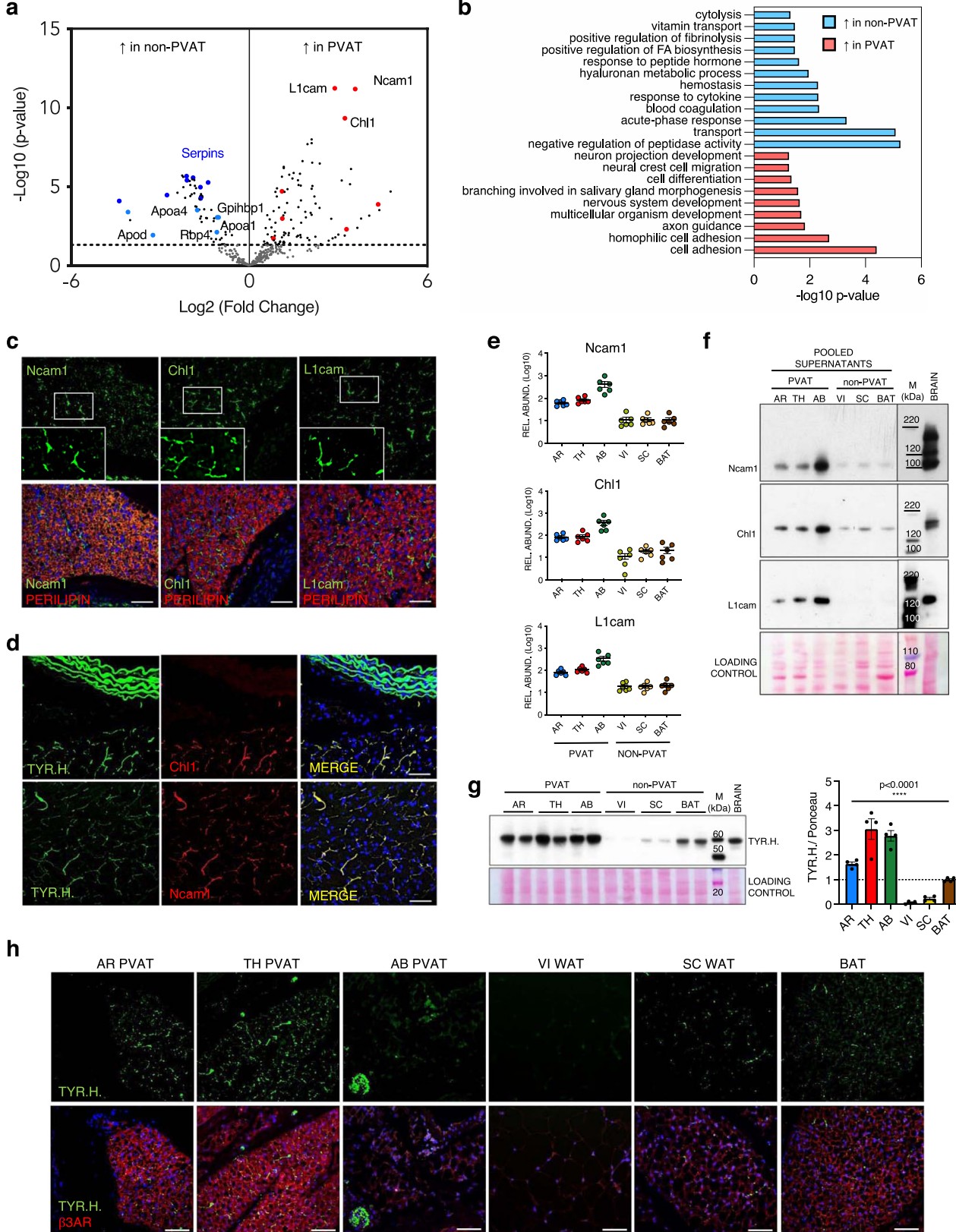

## Negr1 levels and secretion is reduced in the AT of obese mice

Finally, we aimed at identifying molecular factors responsible for the loss of sympathetic innervation in AT depots during obesity. Among the most downregulated proteins in the secretomes of PVAT from *ob/ob* mice was Negr1 (Figs. 3d and 6a and validation on PVAT pooled secretomes in 6b, representative MS/MS spectrum shown in

Supplementary Fig. 4c). Negr1 is a cell-adhesion molecule belonging to the IgLON superfamily that has been reported to modulate cortical and hippocampal neurogenesis[33,34] and to act as a soluble factor enhancing the axonal growth of cortical neurons in vitro[35,36]. While other cell-adhesion molecules involved in axonal development are expressed by sympathetic neurons (Fig. 2c, d and Supplementary Fig. 4d),

**Fig. 2 | Identification of a perivascular secretory signature and characterisation of AT innervation. a** Volcano plot showing proteins differentially secreted by perivascular (AR, TH and AB PVAT, $N = 18$) and non-perivascular (VI and SC WAT and BAT: non-PVAT, $N = 18$) AT depots ($N = 6$ biological replicates per depot). The limma package was used to compare different groups using the Ebayes algorithm (paired analysis) followed by Benjamini–Hochberg adjustment. Light-blue spots: proteins involved in lipid transport. Dark blue spots: serpin family members. Red spots: proteins involved in axon guidance and nervous system development. **b** Gene Ontology analysis showing Biological Processes categories enriched in non-PVAT and PVAT secretomes. Fisher's Exact test p-values are shown. **c** Representative immunofluorescence images (from 1 out of 2 replicates) of TH PVAT sections stained for cell-adhesion molecules Ncam1, Chl1 and L1cam. Adipocytes are stained by perilipin. Nuclei are counterstained with DAPI. Insets show higher magnifications of white framed areas. Scale bars: 50 µm. **d** Representative immunofluorescence images (from 1 out of 2 replicates) showing the co-localisation of cell-adhesion

molecules Chl1 and Ncam1 with TYR.H. on TH PVAT sections. Nuclei are counterstained with DAPI. Scale bars: 50 µm. **e** MS-determined relative abundances of cell-adhesion molecules Ncam1, Chl1 and L1cam in different AT depots secretomes ($N = 6$ biological replicates per depot, mean ± SEM are shown). **f** Western blot on pooled AT secretomes stained for cell-adhesion proteins Ncam1, Chl1 and L1cam ($N = 6$ biological replicates per pool). **g** Western blot of representative AT lysates stained for the sympathetic marker TYR.H. and relative quantification ($N = 4$ biological replicates per depot, mean ± SEM, one-way Anova, $p = 2.83E-09$). **h** Representative immunofluorescence images (from 1 out of 2 replicates) showing TYR.H. and β3AR staining on different AT depots sections. Nuclei are counterstained with DAPI. Scale bars: 50 µm. AR aortic arch PVAT, TH thoracic PVAT, AB abdominal PVAT, VI visceral AT, SC subcutaneous AT, BAT interscapular brown AT, Ncam1 neural cell-adhesion molecule 1, Chl1 close homologue of L1, L1cam neural cell-adhesion molecule L1, TYR.H. tyrosine hydroxylase, β3AR β3-adrenoreceptor, M molecular weight marker. Source data are provided as a Source Data file.

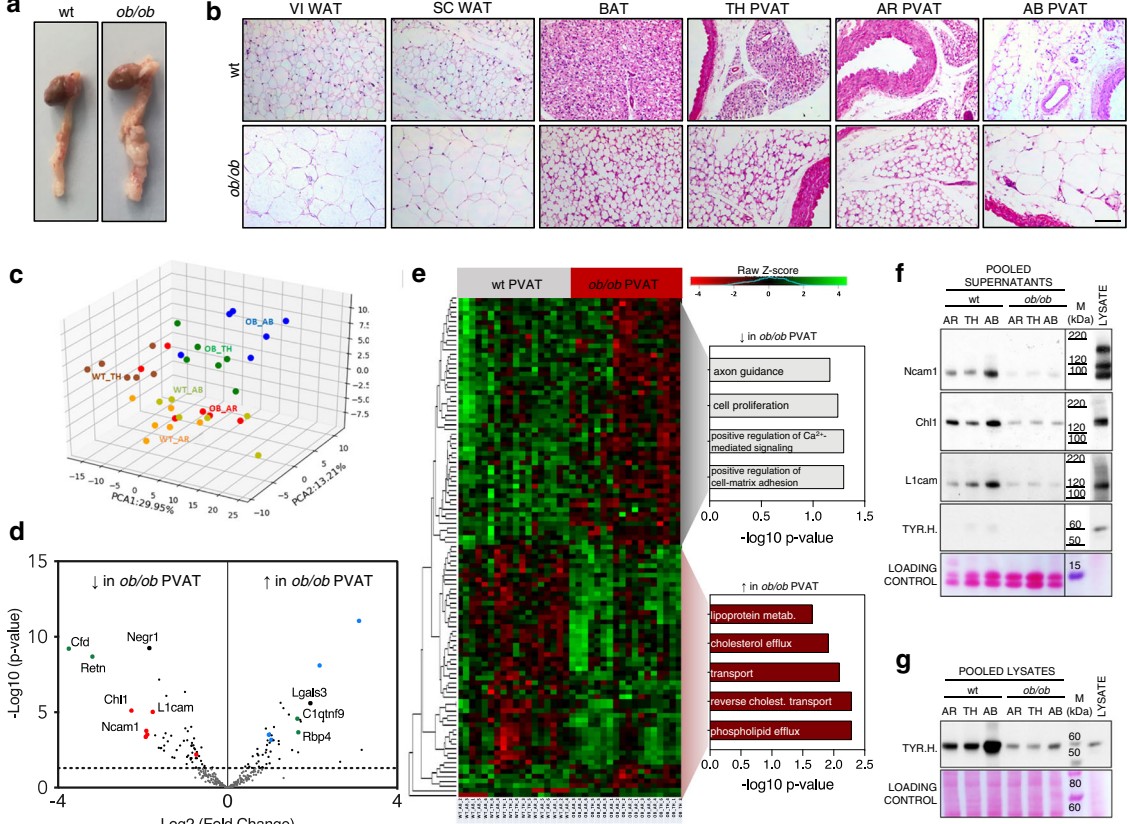

**Fig. 3 | Obesity-induced alterations of PVAT phenotype and secretory profile. a** Representative pictures of heart, aorta and PVAT dissected from wt and *ob/ob* mice. **b** H&E stainings (from 1 out of 3 replicates) on wt and *ob/ob* AT depots sections. Scale bar: 100 µm. **c** Principal Component Analysis of the 293 secreted proteins consistently detected by MS in the conditioned media of wt and *ob/ob* PVAT explants. **d** Volcano plot showing proteins differentially secreted by wt and *ob/ob* PVAT samples ($N = 18$ vs. 18, with $N = 6$ biological replicates per depot). The limma package was used to compare different groups using the Ebayes algorithm (unpaired analysis) followed by Benjamini–Hochberg adjustment. Green spots: well-established adipokines. Blue spots: proteins involved in lipid transport. Red spots: proteins involved in axonal guidance. **e** Heatmap showing the 107 secreted proteins significantly up- or downregulated in the secretomes of *ob/ob* PVAT explants with Gene Ontology analysis showing the Biological Processes categories

enriched in the two groups. Fisher's Exact test p-values are shown. Hierarchical clustering was performed using the complete agglomeration method, Z-score-scaled abundances are displayed. $N = 6$ biological replicates per group. **f** and **g** Western blot on wt and *ob/ob* PVAT pooled conditioned media (**f**) and corresponding pooled lysates (**g**) ($N = 5$ biological replicates per pool) stained for cell-adhesion molecules Ncam1, Chl1 and L1cam and for the intracellular sympathetic marker TYR.H. AR PVAT aortic arch perivascular AT, TH PVAT thoracic PVAT, AB PVAT abdominal PVAT, VI WAT visceral white AT, SC WAT subcutaneous WAT, BAT interscapular brown AT, TYR.H. tyrosine hydroxylase, Cfd adipsin, Retn resistin, Negr1 neuronal growth regulator 1, Ncam1 neural cell-adhesion molecule 1, Chl1 close homologue of L1, L1cam neural cell-adhesion molecule L1, Lgals3 galectin 3, C1qtnf9 complement C1q and tumour necrosis factor-related protein 9, Rbp4 retinol-binding protein 4, M molecular weight marker.

immunostainings on PVAT sections revealed that Negr1 is mainly detected on the plasma membrane of adipocytes and in the extracellular space (Fig. 6c). This is further confirmed by digestion and fractionation of AT explants. As shown in Fig. 6d, Negr1 is only detected

in isolated floating adipocytes (expressing the adipocyte-specific marker perilipin) and not in the stromal vascular fraction (SVF). Conversely, Negr1 protein is completely undetectable in primary sympathetic neurons (Fig. 6e), that on the other do express the sympathetic

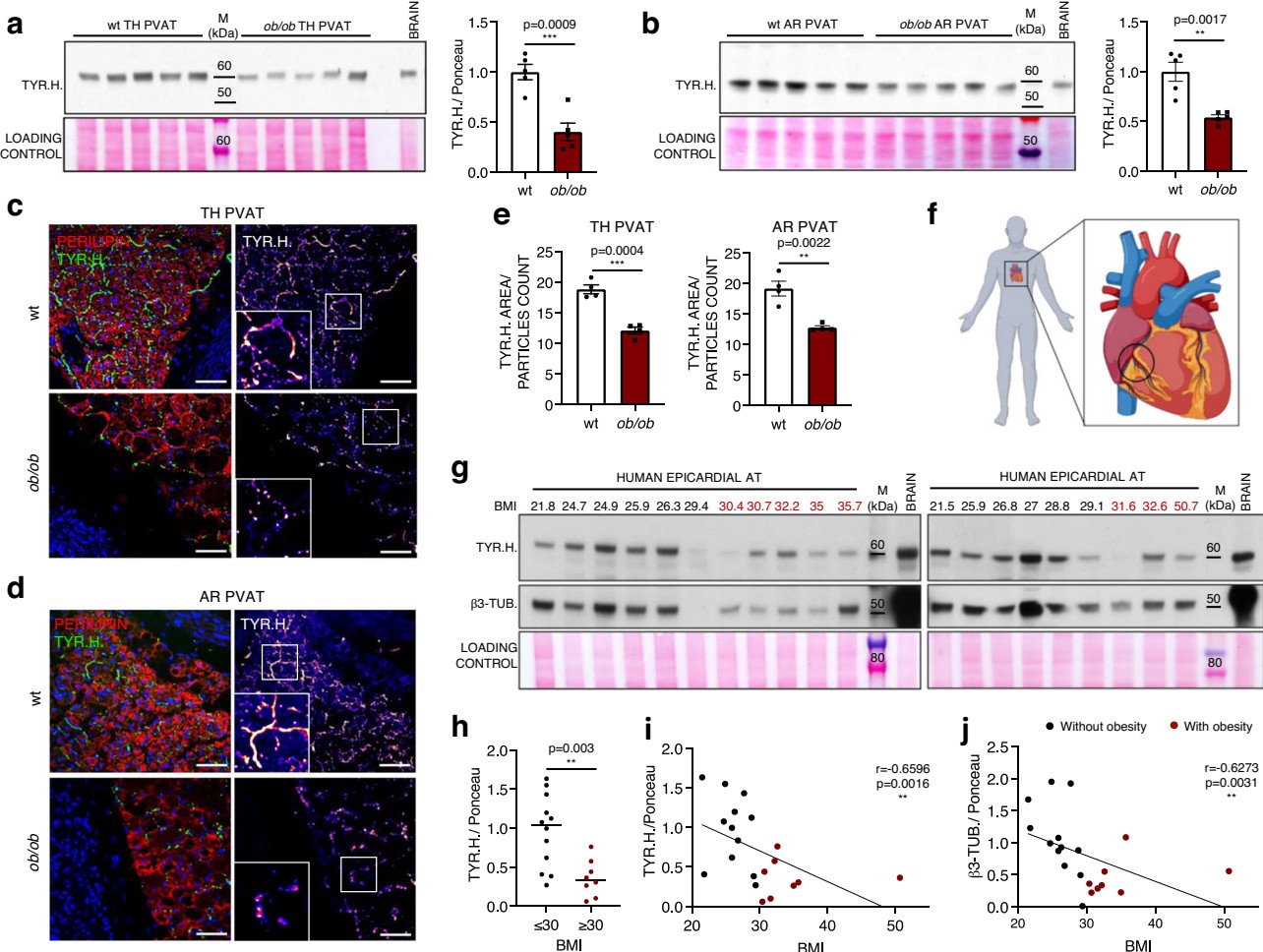

**Fig. 4 | Sympathetic innervation of murine and human AT depots in obesity.**
**a**, **b** Western blots showing TYR.H. abundances in wt and *ob/ob* TH (**a**) and AR (**b**) PVAT lysates and relative quantifications. *N* = 5 biological replicates per group, unpaired, two-tailed independent *t*-test (unequal distribution). Bar charts show mean ± SEM. **c**, **d** Representative immunofluorescence images showing TYR.H. distribution (in green and fire) in wt and *ob/ob* TH (**c**) and AR (**d**) PVAT sections. Adipocytes are labelled by perilipin (in red), nuclei are counterstained with DAPI. Insets show higher magnifications of framed areas. Scale bar: 50 μm. **e** Bar charts showing the average size of TYR.H. positive particles (mean ± SEM) obtained from 4 mice per condition, with 5 to 10 sections analysed in each mouse. *N* = 4, unpaired two-tailed independent *t*-test (unequal distribution). **f** Schematic cartoon showing the localisation of the human epicardial fat collected for this study. Created with BioRender.com. **g** Western blot showing TYR.H. and β3-TUB. staining on human epicardial fat lysates from 12 patients without (BMI ≤ 30, in black) and 8 patients with (BMI ≥ 30, in red) obesity. **h** Scatter dot plot showing the quantification of TYR.H. in epicardial fat samples from patients without (*N* = 12) and with (*N* = 8) obesity. Single values and median are shown, unpaired two-tailed independent *t*-test (unequal distribution). **i**, **j** Linear regression analysis showing the correlation between TYR.H. (**i**) or β3-TUB. (**j**) levels assessed by Western blot and BMI. *N* = 20, Spearman correlation analysis, two-tailed *p*-value. TH thoracic PVAT, AR aortic arch PVAT, TYR.H. tyrosine hydroxylase, BMI body mass index, β3-TUB β3-tubulin, M molecular weight marker. Source data are provided as a Source Data file.

marker tyrosine hydroxylase and other neuronal cell-adhesion molecules previously validated in vivo (Supplementary Fig. 4d). The presence of Negr1 in AT-derived conditioned media is attributed to a matrix metalloproteases (MMP)-mediated cleavage, as incubation of AR and TH PVAT explants with the broad-spectrum MMP inhibitor GM6001 abolishes Negr1 detection in their secretomes (Fig. 6f). This result with GM6001 is replicated in SC WAT (Supplementary Fig. 4e). In contrast, GM6001 does not affect the secretion of other adipokines such as plasminogen activator inhibitor-1 and adiponectin (Fig. 6f and Supplementary Fig. 4e). The decreased abundance of Negr1 in *ob/ob* adipose secretomes extends to all AT depots (Supplementary Fig. 4f) and was validated for SC WAT (Supplementary Fig. 4g). Negr1 is significantly downregulated in TH and AR PVAT lysates as well as SC WAT lysates from *ob/ob* mice compared to wt (Fig. 6g, h and Supplementary Fig. 4h). A significant decrease of Negr1 is also detected in AR PVAT and SC WAT lysates from DIO mice compared to lean controls (Supplementary Fig. 4i), while a higher variability was observed in the TH PVAT from DIO mice compared to *ob/ob* mice.

## Negr1 exerts a trophic effect on sympathetic neurons

Upon co-staining with tyrosine hydroxylase in PVAT sections, we observed that in wt mice Negr1 accumulates in proximity of sympathetic neurites, whereas this juxtaposition is lost in *ob/ob* mice (Fig. 7a), suggesting that Negr1 may play an important role in the guidance and maintenance of terminal neuronal arborisations.

To investigate whether soluble Negr1 is indeed endowed with neurotrophic functions, we performed a dose-finding experiment culturing primary sympathetic neurons at very low density for 24 h in presence of 1 ng/ml NGF (which is essential for neuronal growth and survival) and increasing concentrations of soluble recombinant Negr1 (0, 1, 10, 100, 500, 1000 ng/ml). As shown in Supplementary Fig. 5a, Negr1 addition resulted in an enhanced axonal elongation, with the lowest dose found to be significantly effective being 500 ng/ml. Additional experiments showed that the same dose of Negr1 significantly increased the total axonal length and the number of branching points in presence of both 1 and 10 ng/ml NGF, while it had no effect on the average number of principal neurites emerging from

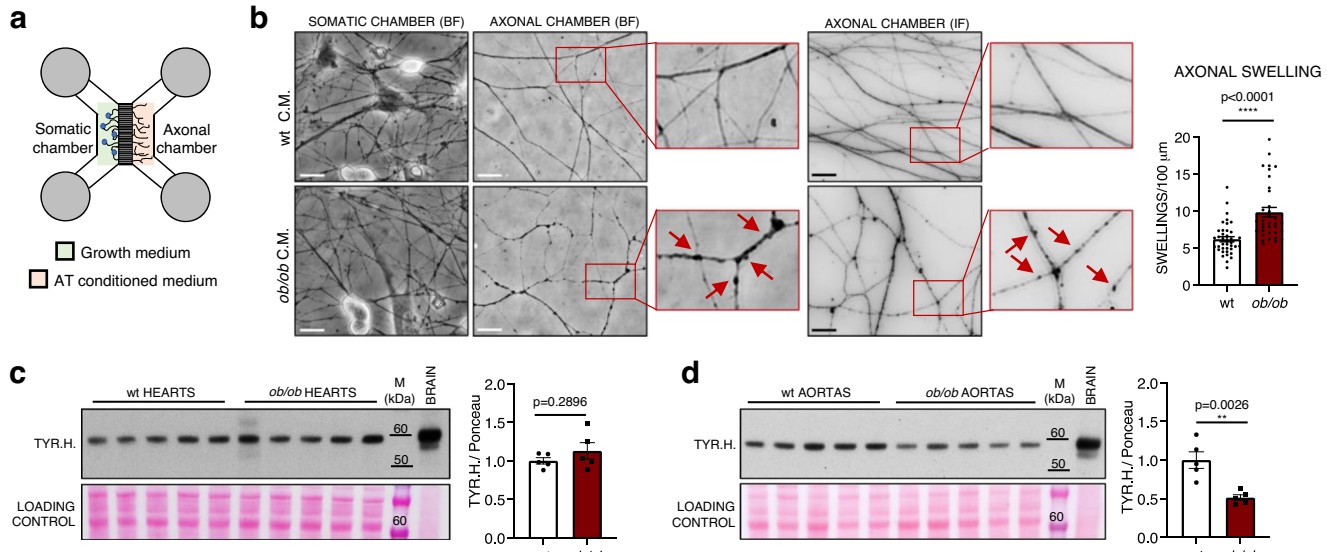

**Fig. 5 | Local effect of AT-conditioned media on cultured sympathetic neurons.** **a** Schematic cartoon showing the structure of microfluidic devices employed for sympathetic neuronal cultures. **b** Representative images of SCG sympathetic neurons seeded on microfluidic devices, with the axonal chamber incubated for 24 h with either wt or *ob/ob* fat explants conditioned media (C.M.) showing brightfield appearance (BF, left panel) and β3-tubulin staining (IF, right panel) of sympathetic neurites. Arrows point at axonal swellings. Scale bars: 20 μm. Bar chart shows the quantification of axonal swellings normalised on total axonal length. Data are shown as mean ± SEM of 39 and 35 random fields acquired from 3 independent experiments and analysed by unpaired two-tailed independent *t*-test (unequal distribution), *p* = 2.98E-06. **c**, **d** Western blots showing TYR.H. abundances in wt and *ob/ob* hearts (**c**) and PVAT-denuded aortas (**d**) lysates and relative quantifications. *N* = 5 biological replicates, unpaired two-tailed independent *t*-test (unequal distribution). Data are shown as mean ± SEM. SCG superior cervical ganglia, TH thoracic PVAT, AR aortic arch PVAT, TYR.H. tyrosine hydroxylase, M molecular weight marker. Source data are provided as a Source Data file.

the neuronal somata (Fig. 7b, c). This indicates that Negr1 significantly enhances the growth and arborisation of distal sympathetic neurites. Addition of Negr1 also attenuated the neurodegenerative effect of *ob/ob* adipose conditioned media on sympathetic axons, as evidenced by the significant reduction of axonal swellings that developed over the time course of incubation (Fig. 7d, e and lower magnification of axonal β3-tubulin staining in Supplementary Fig. 5b). Finally, we decided to test the effect of Negr1 on adipose-sympathetic innervation in vivo. Due to the poor accessibility of PVAT, we focussed on the SC WAT, which is also characterised by decreased innervation and Negr1 secretion in *ob/ob* mice (see Supplementary Figs. 3a and 4g). Subcutaneous osmotic minipumps connected to a catheter delivering recombinant Negr1 (1 μg/day) were implanted at the level of the inguinal SC WAT of *ob/ob* mice (see Fig. 7f). Local protein abundances of Negr1 in the SC AT were found to be significantly increased with respect to mice only receiving saline, while no changes were observed in the same tissues for Negr1 mRNA (Supplementary Fig. 5c, d). Negr1 protein levels were unchanged in other distal AT depots (see Supplementary Fig. 5c). As shown in Fig. 7g, h, the local delivery of Negr1 for 2 weeks did not induce any significant change in the body weight of mice, nor in the weight of the SC WAT depot. However, a significant increase of tyrosine hydroxylase was observed in the SC WAT lysates of *ob/ob* mice upon administration of Negr1 and was replicated for the axonal marker β3-tubulin (Fig. 7i), demonstrating that Negr1 promotes sympathetic axonal growth in vivo.

## Discussion

This proteomics study defined the secretory profiles of different AT depots and revealed a denser sympathetic innervation of the PVAT compared to other fat tissues. Moreover, Negr1 was identified as a neurotrophic factor that is produced by adipocytes. In *ob/ob* mice, we observed a loss of sympathetic axons in perivascular, SC and BAT depots. Mechanistically, this neurodegeneration is due, at least in part, to the downregulation of Negr1 by obese adipocytes. Administration of Negr1 partially restored the impaired neurotrophic effects of

AT-conditioned media from *ob/ob* mice in vitro and promoted sympathetic axonal growth in vivo. Our results present a molecular mechanism for an impaired adipose-neuronal crosstalk in obesity.

In recent years, the importance of AT as an endocrine organ has been widely recognised, with a rising number of adipokines being identified as pathophysiological regulators[8,10,11]. Few proteomics studies have been conducted to characterise the secretome of canonical fat depots[37–39], but none to profile the PVAT. The PVAT is unique with its role in regulating vascular functions and reflecting vascular inflammation[12–15,19,40,41]. In the present work, we exploited proteomics to comprehensively characterise the secretory profile of PVAT in wt and *ob/ob* mice. Peri-aortic fat was collected from 3 areas (AR, TH and AB aorta) to account for the phenotypical differences observed across these distinct vascular regions. Proteomics analysis on conditioned media of 3 perivascular and 3 canonical fat depots (VI and SC WAT, BAT) led to the identification of 526 secreted proteins. These secreted proteins are either endowed with a signal peptide for secretion (SignalP[22]) or listed in the Matrisome database[23]. While our analysis achieved a much greater proteome coverage than the previous publications to date[37–39], another source of secreted proteins are extracellular vesicles and exosomes, and additional studies will be required to specifically profile their protein content. As expected, one of the strongest determinants of the secretory profile of each depot is its WAT or BAT phenotype. In agreement with previous studies[9,38], VI and SC WATs exhibit a higher secretory capacity, predominantly of proteins involved innate immune response and lipid metabolism. Among BAT-enriched factors we detected Kallikrein-1, S100b and Visfatin, known to be implicated in the regulation of browning[26], BAT innervation[25] and insulin secretion[24].

Proteins involved in axonal guidance and nervous system development were surprisingly more abundant in the secretome of PVAT compared to non-PVAT. The most upregulated ones were cell-adhesion molecules L1cam, Ncam1 and Chl1[42], expressed by tyrosine hydroxylase-positive axons embedded within the adipose parenchyma. Their higher abundances in PVAT secretomes reflect a

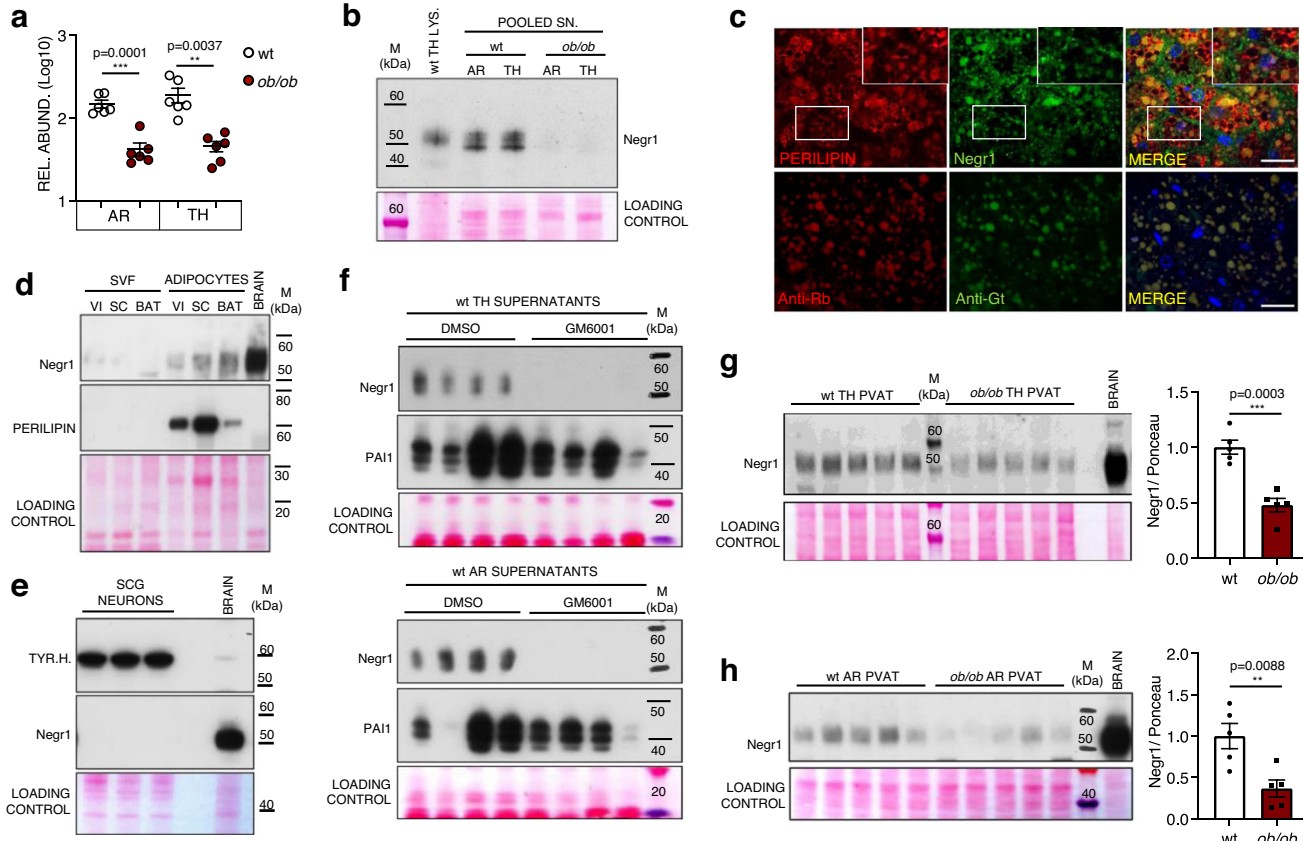

**Fig. 6 | Negr1 levels and secretion are reduced in the PVAT of *ob/ob* mice.**
**a** MS-determined relative abundances of Negr1 in the secretomes of wt and *ob/ob* AR and TH PVAT. *N* = 6 biological replicates per group, unpaired, two-tailed *t*-test, mean ± SEM is shown. **b** Western blot on wt TH PVAT lysate (LYS.) and supernatants (SN.) pools (*N* = 6 biological replicates per pool) stained for Negr1. **c** Representative immunofluorescence images (1 out of 2 replicates) showing Negr1 localisation in wt TH PVAT. Adipocytes are labelled by perilipin. Nuclei are counterstained with DAPI. Insets show higher magnification of white framed areas. Scale bar: 50 μm. **d** Western blot showing Negr1 and perilipin levels in stromal vascular fraction (SVF) and adipocytes obtained after digestion and fractionation of VI and SC WAT and BAT explants from wt mice (1 out of 2 replicates). **e** Western blot on primary murine sympathetic neurons lysates from 3 biological replicates stained for TYR.H. and

Negr1. **f** Western blot on supernatants from TH (top) and AR (bottom) PVAT explants from wt mice incubated for 24 h in absence (DMSO) or presence of the wide spectrum MMP inhibitor GM6001 and stained for Negr1 and PAI1. *N* = 4 biological replicates per group. **g** and **h** Western blots showing Negr1 abundances in wt and *ob/ob* TH (**g**) and AR (**h**) PVAT lysates and relative quantifications. *N* = 5 biological replicates, unpaired two-tailed independent *t*-test (unequal distribution). Data are shown as mean ± SEM. AR aortic arch PVAT, TH thoracic PVAT, VI visceral AT, SC subcutaneous AT, BAT interscapular brown AT, TYR.H. tyrosine hydroxylase, Negr1 neuronal growth regulator 1, PAI1 plasminogen activator inhibitor-1, SCG superior cervical ganglia, M molecular weight marker. Source data are provided as a Source Data file.

greater content of sympathetic neurites, with AR and TH PVAT being more densely innervated than all other investigated ATs. β3-adrenoreceptors are also highly expressed on the surface of adipocytes in AR and TH PVAT[2,4].

In *ob/ob* mice, PVAT remodelling is evident by the presence of hypertrophic intracellular lipid droplets both in white and brown adipocytes. By analysing the conditioned media of PVAT explants, we demonstrated that obesity induces substantial alterations in their secretory profiles. In addition to higher secretion of proteins involved in lipid transport and metabolism, the increase of galectin 3 (Lgals3, Fig. 3d) is consistent with an augmented infiltration of macrophages in the adipose parenchyma, which promotes chronic inflammation and insulin resistance[43,44]. Additionally, some well-established adipokines— including adipsin, retinol-binding protein 4, complement C1q and tumour necrosis factor-related protein 9 and resistin—were found dysregulated in *ob/ob* mice. Unexpectedly, the neuronal cell-adhesion molecules L1cam, Chl1 and Ncam1 were among the most downregulated proteins in *ob/ob* PVAT secretomes, and their decrease reflects a reduction of intra-parenchymal sympathetic fibres in the PVAT of *ob/ob* mice. Of note, a similar observation was also made in SC WAT and in BAT, confirming a recent report by Wang and colleagues[30].

Intriguingly, impaired innervation is also apparent in the epicardial fat of patients affected by obesity, suggesting that the observations in *ob/ob* mice also extend to humans.

While Wang and colleagues elegantly demonstrated a central nervous system-mediated mechanism that links systemic leptin resistance in obesity to SC WAT and BAT denervation in mice[30], additional regulatory processes may also modulate innervation at a local level[25]. Indeed, fat-secreted factors have previously been implicated in the paracrine regulation of intra-adipose-sympathetic plasticity in the context of cold-induced beiging[45,46]. Here, we found that incubation of primary neurons with conditioned media of fat explants from *ob/ob* mice induces the formation of axonal swellings, a well-established hallmark of axonal degeneration[32]. Tyrosine hydroxylase was significantly reduced in aortas—but not hearts—from *ob/ob* mice, further supporting the concept that neurodegeneration is a consequence of the local alterations affecting the perivascular environment during obesity and impacting on the nearby vascular wall.

When we searched the PVAT secretomes for changes that may be responsible for this loss of sympathetic neurites, adipocyte-derived Negr1, a IgLON cell-adhesion molecule, was found to be significantly

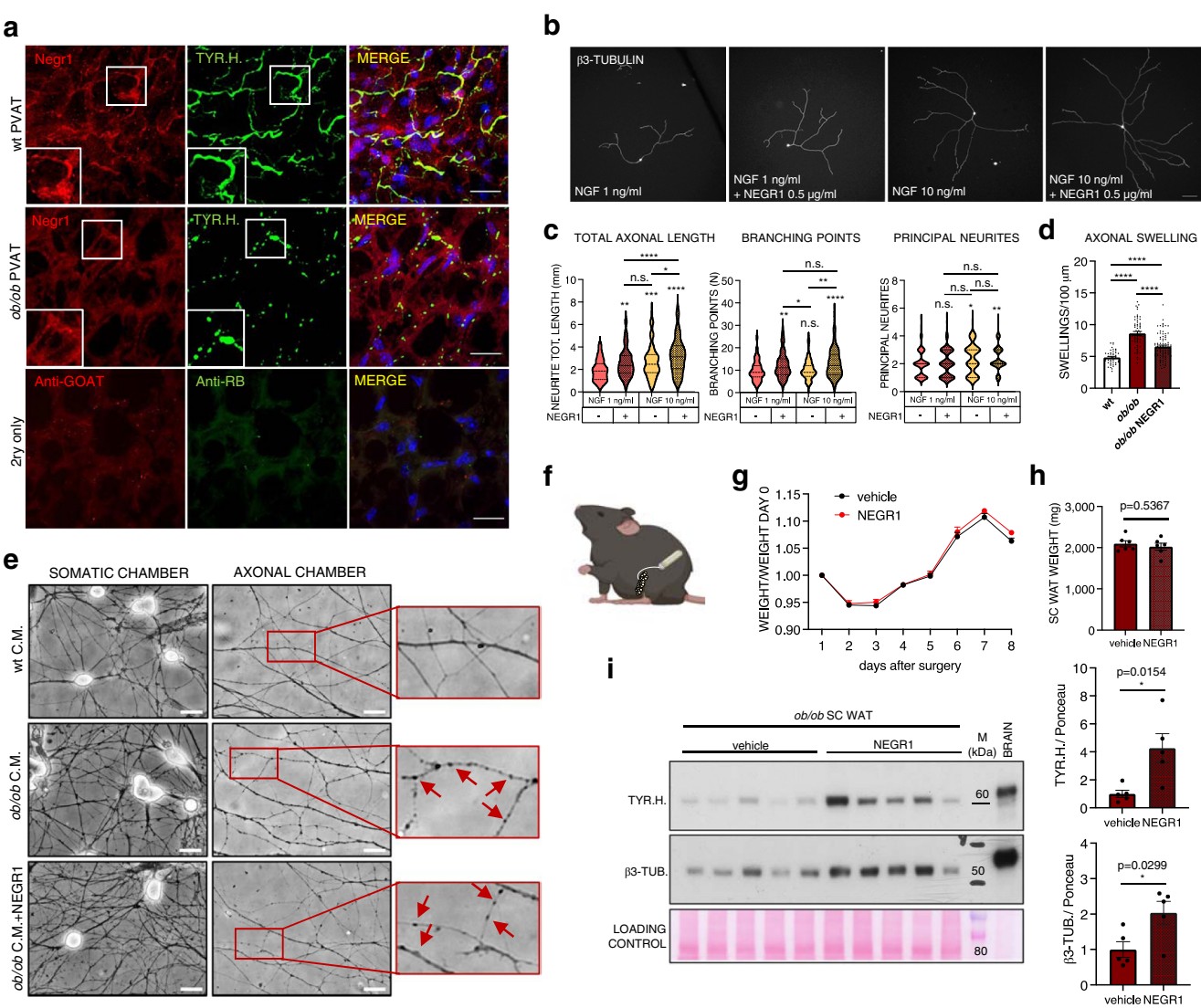

**Fig. 7 | Negr1 exerts a trophic effect on sympathetic neurons.**
**a** Immunofluoescence images showing Negr1 and TYR.H. distribution in wt and *ob/ob* TH PVAT sections (from 1 out of 2 replicates). Insets show higher magnification of framed areas. Nuclei are counterstained with DAPI. Scale bar: 50 μm. **b** Representative micrographs of SCG sympathetic neurons cultured at low density for 24 h in presence of 1 or 10 ng/ml NGF, with or without 500 ng/ml recombinant Negr1, and stained for β3-tubulin. Scale bar: 100 μm. **c** Quantifications of total axonal length, branching points and principal neurites on primary sympathetic neurons incubated for 24 h in medium supplemented with the indicated factors. Violin plots show median and quartiles of 80 < N < 120 neurons per condition from 3 independent experiments. *$p < 0.05$; **$p < 0.01$; ***$p < 0.001$; ****$p < 0.0001$ by one-way ANOVA followed by Tuckey's multiple comparisons test. **d** Bar chart shows the quantification of axonal swellings normalised on total axonal length. Data are shown as mean ± SEM of 40 to 80 random fields from 3 independent experiments analysed by one-way ANOVA followed by Tuckey's multiple

comparison test. ****$p < 0.0001$. **e** Representative live BF images of SCG sympathetic neurons seeded on microfluidic devices, with the axonal chamber incubated for 24 h with conditioned media (C.M.) from wt and *ob/ob* fat explants +/− recombinant Negr1 (500 ng/ml). Arrows point at axonal swellings. Scale bars: 20 μm. **f** Localisation of catheter-connected osmotic minipumps implanted subcutaneously in *ob/ob* mice. Created with BioRender.com. **g** Trend of *ob/ob* mice weight from the surgery day until mice were culled. **h** Bar graph showing the weight of SC WAT pads on day 14 after minipumps implantation. **i** Western blot showing TYR.H. and β3-TUB. abundances in the SC WAT of *ob/ob* mice implanted with subcutaneous minipumps delivering either saline (vehicle) or recombinant Negr1 and relative quantification. N = 5 biological replicates, unpaired two-tailed independent t-test (unequal distribution). Data are shown as mean ± SEM. TH thoracic PVAT, NGF nerve growth factor, Negr1 neuronal growth regulator 1, TYR.H. tyrosine hydroxylase, β3-TUB β3-tubulin, SCG superior cervical ganglia, M molecular weight marker. Source data are provided as a Source Data file.

decreased in *ob/ob* mice. Negr1 was reported to also act as a soluble factor, promoting the growth and arborisation of cortical neurites[35,36]. However, no information is available on potential effects of Negr1 on sympathetic neurons. Here, we demonstrated that the addition of recombinant Negr1 significantly increases the arbour complexity of primary sympathetic axons, enhancing their elongation and branching. Supplementation of recombinant Negr1 attenuates the neurodegenerative effect of PVAT conditioned media from *ob/ob* mice on sympathetic axons. Finally, the local delivery of Negr1 in the SC WAT of *ob/ob* mice leads to an overall increase of sympathetic innervation, as

evidenced by the increased tyrosine hydroxylase content of SC WAT lysates from Negr1-treated mice. Taken together, these data suggest that Negr1 is involved in the maintenance of intra-parenchymal neurites, with a reduced production by obese adipocytes resulting in a lack of trophic support that contributes to the degeneration of sympathetic terminals in obesity. Notably, reduced levels of adipose-sympathetic innervation and Negr1 abundance are also observed in DIO mice, although to a lesser extent than in *ob/ob* animals, as the DIO mouse is a milder model of obesity. Of note, generalisability to female animals was not investigated and could represent a limitation of the current

study. Male and female patients, however, were included in the study of epicardial tissues.

In the past decade, Negr1 has gained increasing attention since it was identified as new locus associated with human obesity by three independent genome-wide association studies[47–49]. Besides defective neurogenesis and altered behaviour[33,34], mice lacking Negr1 display increased adiposity and adipocyte hypertrophy[50]. A regulatory role has been proposed for Negr1 in the modulation of intracellular cholesterol trafficking[51], but the mechanistic link responsible for the phenotype of Negr1 null mice has not been fully elucidated yet. The trophic effect exerted by Negr1 on sympathetic neurites represents a promising mechanism for the impairment of energy metabolism associated with AT hypertrophy and obesity.

In summary, we demonstrated that obesity is associated with profound changes in the secretomes of PVAT, that contribute to the loss of sympathetic neurites observed within the adipose parenchyma and in the nearby vascular wall. Decreased innervation is also observed in SC WAT and BAT of *ob/ob* mice, in ATs from DIO animals, and in the epicardial fat of human patients affected by obesity. As AT-embedded sympathetic inputs are key drivers of browning[4] and thermogenesis[2], a dysregulated crosstalk between the SNS and AT is likely to contribute to the energy imbalance characterising the progression of metabolic diseases. The present study identifies Negr1 as an important mediator in the communication between AT and the SNS, as supplementation of soluble Negr1 restores the trophic effect of AT secretomes from *ob/ob* mice on sympathetic neurons. Unravelling molecular mechanisms mediating the SNS-AT crosstalk provides opportunities for the development of novel therapeutic strategies.

## Methods

### AT collection from mice
All procedures and animal welfare were in accordance with either: (i) the Guidance on the Operation of the Animals (Scientific Procedures) Act, 1986 (United Kingdom) with ethical approval obtained from the King's College London Committee for the Review of Ethics and Welfare (Licence number: 70/0843, Prof. Qingbo Xu and PP2172591, Dr. Ursula Mayr, approved by the Home Office under the Establishment license number X24D82DFF held by King's College London) or (for HFD experiments) (ii) the guidelines for the use of experimental animals as given by "Deutsches Tierschutzgesetz" with ethical approval from the local Research Board for animal experimentation (State Agency for Nature, Environment and Consumer Protection, file ref. 81-02.04.2017.A458). Wt male C57BL/6J mice and age-matched *ob/ob* animals with the same genetic background (B6.Cg-*Lep^ob*/J) were purchased from Charles River. 18 weeks old wt and *ob/ob* mice were used for AT harvesting and secretomes collection. Although wt and *ob/ob* age-matched mice employed here were not littermates, this is in line with many published articles[30,52], and we expect this not to represent a significant limitation for this study. AT samples from DIO mice and matched controls were kindly provided by Prof. Maria Grandoch (Institute of Translational Pharmacology, Düsseldorf, Germany). C57BL/6J mice were purchased from Janvier Labs and fed with a 60% HFD (Ssniff #S7200-E010, modified after Bio Serv #F1850) from the age of 5 weeks for 16 weeks while in parallel control mice were fed with a matched low-fat diet (Ssniff #E157453-04). All animals were housed in 12 h controlled dark/light cycle, under constant temperature (21 °C) and humidity (55%), with free access to food and water. Mice were checked daily by animal facility qualified staff to ensure that animal welfare was safeguarded during regular housing and experiments. For AT collection, animals were euthanised by anaesthesia overdose, and the following AT samples were dissected: VI WAT (located inside the peritoneal cavity around the internal organs), inguinal SC WAT (underneath the skin) and interscapular BAT. PVAT was collected from around three distinct regions of the murine aorta: the aortic arch (AR PVAT), the thoracic aorta (TH PVAT, from underneath aortic arch to diaphragm) and the abdominal aorta (AB PVAT, from underneath the diaphragm to infra-renal branch). All tissues were carefully excised and transported from the animal house operating room to the laboratory in sterile DMEM/F12 medium (ThermoFisher Scientific).

### In vivo osmotic minipumps implantation
Osmotic minipumps (Alzet, model 1004) were filled with a sterile saline solution containing murine recombinant Negr1 (Sino Biological, delivery rate: 1 µg/day) or saline alone and connected to a polyvinylchloride catheter according to manufacturer's instructions. Proper functioning of the minipumps and delivery of solution through the catheter was assessed during the priming phase (as per manufacturer's instructions). Ten weeks old male *ob/ob* mice were anaesthetised by intraperitoneal (i.p.) injection of ketamine (75 mg/kg) and medetomidine (1 mg/kg), an incision was made on the left side of the lower back (above the hip) and a subcutaneous pocket was created. The minipump was located inside the subcutaneous pocket and the connected catheter's end was inserted in the left SC WAT depot. The wound was then closed with an interrupted suture using 6-0 vicryl thread and antisedan (5 mg/kg) was i.p. administered to the mice. Buprenorphine and non-steroidal anti-inflammatory drugs were administered to the mice immediately before the surgery to minimise discomfort. All animals were monitored daily and weighed every other day after surgery. After 2 weeks, mice were euthanised by anaesthesia overdose. After minipump and catheter removal, the left SC WAT pad was collected, weighed, extensively washed and snap-frozen for subsequent analysis. Leftover volumes inside minipumps were measured to further confirm that the expected amount of saline/Negr1 had been delivered during the time course of the in vivo experiment.

### Human epicardial AT samples
Epicardial AT samples were collected from patients undergoing isolated coronary artery bypass graft surgery (CABG, $N = 11$) or aortic valve replacement (AVR, N = 9, see Supplementary Table 1) at St. George's Hospital in London (UK) in 2012-2013 (Research Ethic Committee reference for ethical approval: 12/LO/0422 obtained from National Research Ethics Service Committee London–Bromley; St. George's R&D approval: 12.0019). All study participants gave written informed consent before taking part in the study. Participants did not receive any compensation. Male and female patients were included in the study, with ages ranging from 36 to 85 years (see Source data file and Supplementary Fig. 3e for more detailed information). Exclusion criteria included: not adequate understanding of the study and consent; antiarrhythmic therapy other than beta blockers; presence of temporary or permanent pacemaker; off-pump-CABG surgery; systemic inflammatory disease requiring immunomodulating therapy; recent steroid therapy; HIV infection; lipodystrophy. Details on collection were also described previously[53]. According to clinical needs, approximately 500 mg of tissue were most commonly harvested from the area around the right coronary artery on the right ventricle. The samples were immediately transferred to the laboratory, minced, washed 3 times in sterile PBS to eliminate blood contamination and finally lysed in lysis buffer (125 mmol/l Tris-HCl, 2% SDS, pH 7.4) containing proteases- and phosphatases-inhibitors (Roche) and homogenised with beads (Azer Scientific). Protein extraction, quantification and processing for Western blot were performed as described in the next sections for murine AT samples.

### Preparation of AT secretomes
In a biological safety cabinet, excised AT samples were minced in sterile PBS and centrifuged at $250 \times g$ for 10 min at room temperature (RT) in sterile PBS to remove blood contamination, as previously described[38]. After washing, samples were transferred onto

24-well plates and incubated overnight (37 °C, 5% $CO_2$) in phenol red- and serum-free DMEM/F12 medium containing 100 U/ml penicillin and 100 μg/ml streptomycin (ThermoFisher Scientific, incubation volume was adjusted for different AT masses). In some experiments, 50 μM GM6001 (broad-spectrum MMP inhibitor, abcam) dissolved in DMSO−or DMSO alone− was added to the medium for the whole time of incubation. After 24 h, supernatants were collected, centrifuged (18,000 × g, 10 min at 4 °C) to remove cell debris and concentrated using Amicon® Ultra-4 Centrifugal Filter Devices (3 K cut-off, Merck) according to the manufacturer's instructions. Protein concentrations of supernatants were determined using the Pierce BCA™ Protein Assay Kit (ThermoFisher Scientific) according to supplier's instructions.

## AT explants digestion and fractionation

VI and SC WAT and BAT were dissected for wt male mice as described in previous sections. All AT samples were finely minced, extensively washed in sterile PBS and centrifuged at 250 × g for 10 min at room temperature (RT) in sterile PBS to minimise blood contamination. Fat tissues were then placed in a sterile tube containing 1 mg/ml Type II Collagenase (ThermoFisher Scientific) in DMEM/F12 and incubated in an orbital shaker at 180 rpm at 37 °C. After 1 h, tissues were repeatedly re-suspended with a sterile pipette to favour the mechanical dis-aggregation and then centrifuged at 200 × g for 10 min at RT. This led to the formation of two clearly separated fractions: (i) a pellet composed of stromal vascular fraction (SVF) cells and (ii) a floating layer composed of adipocytes. Both fractions were collected separately and lysed. A protein-extraction step (see next section) was performed on lysed adipocytes to reduce lipid contamination. Lysates were then quantified and processed for Western blot.

## Preparation of AT protein lysates

Fat tissue explants were placed into lysis buffer (125 mmol/l Tris-HCl, 2% SDS, pH 7.4) containing proteases- and phosphatases-inhibitors (Roche) and homogenised with beads (Azer Scientific). The obtained lysates were incubated for 1 h at 4 °C under rotation and then centrifuged (14,000 × g, 10 min, 4 °C) to pellet the insoluble cell debris. The upper fatty layer was removed and the clear underlying fraction underwent a further protein-extraction step to minimise lipid contamination[54]. Briefly, 500 μl of tissue lysate were diluted in 1875 μl of a methanol-chloroform (2:1) mixture and after a 15 min incubation on ice, 625 μl of chloroform and 625 μl of water were added to change the water/chloroform/methanol ratio from 0.8:1:2 to 1.8:2:2. After centrifugation (800 × g, 5 min, 4 °C), the protein disk lying between the lower lipid phase and the upper aqueous phase was collected, dissolved in lysis buffer and the protein concentration was determined with the Pierce BCA™ Protein Assay Kit (ThermoFisher Scientific) according to the manufacturer's instruction.

## RNA preparation and quantitative PCR

RNA extraction was performed using the miRNeasy Mini Kit (Qiagen) following the manufacturers' protocol. RNA concentration (Abs 260 nm) and purity (260/280) were measured in 1 μl of eluted RNA using spectrophotometry (NanoDrop ND-1000; Thermo Scientific). The RNA was then reverse transcribed using random hexamers with SuperScript VILO MasterMix (Invitrogen) according to manufacturers' protocol. The reverse transcription reaction was set up by diluting 300 ng of RNA in 8 μl of water and then adding 2 μl of SuperScript VILO MasterMix. The reverse transcription reaction was performed in a Veriti thermal cycler (Applied Biosystem) with the following protocol: incubation at 25 °C for 10 min followed by 2 h incubation at 42 °C. The reaction was terminated by incubation at 85 °C for 5 min. Samples were kept at 4 °C for immediate use. The quantitative PCR was performed using 1.5 ng of reverse transcription product diluted in RNase-free water. TaqMan hydrolysis probes (Mm01317328_m1 catalogue number

4331182 for *Negr1* and Mm01277042_m1 catalogue number 4448489 for *Tbp*, Applied Biosystems) were used for quantitative PCR analysis according to the manufacturer's instructions. Data were normalised on *Tbp* (TATA-box-binding protein) and analysed using ViiA 7 software (Applied Biosystems). Relative amounts of the targets were calculated using the $2^{-\Delta\Delta Cq}$ method, with statistical analysis performed on ΔCq values.

## In-solution digestion of secretome samples

After concentration of conditioned media and protein quantification (described in previous Methods section), 10 μg protein/sample were denatured by addition of urea and thiourea (final conc. 6 M urea, 2 M thiourea), reduced by addition of dithiothreitol (final conc. 10 mM) and incubated for 1 h at 37 °C, 240 rpm. Samples were cooled down to RT prior to alkylation with iodoacetamide (final conc. 50 mM) for 1 h at RT in the dark. Proteins were then precipitated by adding pre-chilled acetone (8:1 acetone:protein volume ratio) and incubated at −20 °C overnight. Subsequently, samples were centrifuged (16,000 × g, 40 min, 0 °C) and supernatants were discarded. The remaining protein pellets were dried using a SpeedVac vacuum concentrator (Thermo Scientific, Savant SPD131DDA) re-suspended in 0.1 M triethylammonium bicarbonate buffer (pH 8.2) containing trypsin (1:50 protease:protein weight) and incubated overnight on a shaker at 37 °C, 240 rpm. The digestion was stopped by acidification with trifluoroacetic acid (TFA, final conc. 1%). Peptide samples were purified using a 96-well Macro Spin-Columns C-18 (Harvard Apparatus, 74-5657) according to the manufacturer's instructions. Eluates were dried using a SpeedVac. Dried peptide samples were dissolved in 0.05% TFA in 2% acetonitrile (ACN) in water.

## LC-MS/MS method for AT-derived secretome samples

Peptides were separated using an UltiMate 3000 RSLCnano system (Thermo Scientific) coupled to an Orbitrap Fusion Lumos Tribrid mass spectrometer (Thermo Scientific). Samples were injected onto a C18 trap cartridge (Thermo Scientific, cat. no. 160454), at a flow rate of 25 μl/min for 3 min, using formic acid (FA, 0.1%) in water. The nanoLC gradient had a flow rate of 0.25 μl/min to separate peptides on the nanoflow reversed phase column (EASY-Spray C18 Reversed Phase HPLC column, 2 μm, 100 Å, 75μm × 50cm; Thermo Scientific, ES803). The column was connected to an EASY-Spray NG source (Thermo Scientific) and kept at 45 °C. Precursor spectra were collected from an Orbitrap mass analyser (resolution of 120,000 at 200 m/z) over the mass-to-charge (m/z) range 350–1500. Data-dependent $MS^2$ spectra of the most abundant precursor ions were obtained after collision-induced dissociation (CID) and analysis in a linear ion trap with Top Speed mode (cycle time 3 s) and dynamic exclusion (duration 60 s) enabled. LC-MS data were acquired using Tune (version 3.0, Thermo Scientific) and Xcalibur (version 4.1, Thermo Scientific).

## Database search and data analysis

Data were searched against a murine database (UniProtKB/Swiss-Prot, version May 2018; 16,970 entries) using Proteome Discoverer (version 2.2.0.388, Thermo Scientific) linked to a Mascot server (version 2.6.0, Matrix Science). The mass tolerance was set at 10 ppm (parts per million) for precursor ions and 0.8 Da (dalton) for fragment ions. Trypsin was used as digesting enzyme with up to two missed cleavages being allowed. Carbamidomethylation of cysteine and oxidation of lysine, methionine and proline were set as dynamic modifications. False-discovery rates (FDRs) were calculated using a target/decoy strategy. Peptide and protein identifications were validated using the Peptide Validator and Protein FDR Validator nodes of Proteome Discoverer, with the target FDR in both cases having been set at 0.01. Results were filtered for High Protein FDR Confidence (as determined by Protein FDR Validator) and a minimum number of unique peptides per protein of 2.

## Histological and immunohistochemical characterisation of AT depots

For histological analysis, VI and SC WAT and interscapular BAT samples were collected as described in the first Materials and Methods section, whereas the entire aorta wrapped in fat was dissected to characterise the PVAT. Samples were fixed overnight in 4% paraformaldehyde in PBS at 4 °C and then processed for either paraffin- or OCT-embedding. In the first case, tissues were dehydrated and embedded in paraffin, followed by micro-sectioning (7 μm sections). After drying at 60 °C overnight, the sections were deparaffinised in xylene, rehydrated by incubating in 100%, 90%, and 70% ethanol and processed for haematoxylin and eosin (H&E) staining. Slices were then dehydrated through incubation in 70%, 90% and 100% ethanol followed by xylene and finally mounted in DPX new non-aqueous mounting medium (Merck). The visualisation was performed with a Leica DM2000 and images acquired with a LAS V4.3 software. For immunohistochemistry, deparaffinised and rehydrated sections were subjected to antigen retrieval in citrate acid buffer (10 mM sodium citrate, pH = 6, 0.05% Tween) at 95 °C for 20 min, blocked for 1 h at RT in 10% donkey serum, 0.2% glycine, 0.5% Triton in PBS and incubated overnight at 4 °C with primary antibodies diluted in blocking solution. After washing with PBS, samples were incubated for 1 h at RT with the corresponding fluorescent secondary antibodies (ThermoFisher) diluted 1:200 in blocking solution, counterstained with DAPI and mounted with VectaMount AQ (Vector Laboratories). OCT-cryosections (16 μm-thick) were blocked, permeabilised and stained as described for paraffin samples. The following primary antibodies were used for immunostainings: Perilipin-1 ab61682, abcam, 1:200; UCP1 ab23841, abcam, 1:100; Ncam1 ab220360, abcam, 1:100; Chl1, AF2147, R&D, 1:100; L1cam ab208155, abcam, 1:100; tyrosine hydroxylase AB152, Merck, 1:100; beta3-adrenergic receptor ab94506, abcam, 1:100; Negr1 AF5384, R&D, 1 μg/ml. The following secondary antibodies were also used: Donkey anti-Goat IgG (H + L) Cross-Adsorbed Secondary Antibody, Alexa Fluor 647, ThermoFisher, A-2147; Donkey anti-Rabbit IgG (H + L) Highly Cross-Adsorbed Secondary Antibody, Alexa Fluor 594, A-21027; Donkey anti-Goat IgG (H + L) Cross-Adsorbed Secondary Antibody, Alexa Fluor 488, A-11055; Donkey anti-Rabbit IgG (H + L) Highly Cross-Adsorbed Secondary Antibody, Alexa Fluor 647, A-31573. Immuno-stained sections were visualised on a Nikon Spinning Disk confocal microscope, pictures were acquired using NIS-elements 4.0 software and processed with ImageJ. For tyrosine hydroxylase quantification, a binary image of the tyrosine hydroxylase staining was created from the z-projection of the raw file. The same threshold parameters were applied to all images analysed. Four wt and 4 *ob/ob* mice were used, with 5 to 10 sections analysed per mouse. The average size of tyrosine hydroxylase-positive particles was used as an index of axonal structural integrity[55].

## SDS-PAGE and western blotting

Equal amounts of AT purified lysates and concentrated supernatants were denatured and reduced by heating (95 °C, 5 min) in Laemmli sample buffer (25 mM Tris pH 6.8, 0.5% SDS, 10% glycerol, 2.5% β-mercaptoethanol, and 0.005% bromophenol blue) prior to separation on a NuPAGE® 4-12% Bis-Tris gel (200 V, 1 h) using NuPAGE® MES SDS running buffer (ThermoFisher Scientific). MagicMark™ XP Western Protein Standard and Novex™ Sharp Pre-stain (ThermoFisher Scientific) were used as molecular weight references. Proteins were blotted (350 mA, 2 h, transfer buffer 25 mM Tris, 192 mM Glycine, 20% methanol) onto a nitrocellulose membrane. After protein transfer, the membrane was blocked with 5% non-fat dry milk dissolved in PBST (PBS 0.01% Tween) and incubated overnight at 4 °C with primary antibodies diluted in blocking solution (Perilipin-1 ab61682, abcam, 1:10,000; UCP1 ab23841, abcam, 1:15,000; Ncam1 ab220360, abcam, 1:1000; Chl1, AF2147, R&D, 1:5000; L1cam ab208155, abcam, 1:1000; tyrosine hydroxylase AB152, Merck, 1:1000; tyrosine hydroxylase

AB1542, Merck, 1:1000; Negr1 AF5384, R&D, 1 μg/ml; beta-actin A1978, Sigma Aldrich, 1:10,000; PBEF/Visfatin, MAB40441, clone # 882104, R&D, 1 μg/ml; Serpin E1/PAI-1 AF3828, R&D, 0.3 μg/ml; beta III Tubulin ab18207, abcam, 1:1000; Adiponectin ab22554, abcam, 1:1000). Thereafter, samples were probed with corresponding secondary HRP-conjugated antibodies (Jackson) diluted 1:5000 in blocking solution (specifically: Peroxidase IgG Fraction Monoclonal Mouse Anti-Goat IgG, light-chain specific, Jackson, 205-032-176, Peroxidase IgG Fraction Monoclonal Mouse Anti-Rabbit IgG, light-chain specific, Jackson, 211-032-171, Peroxidase AffinityPure Goat Anti-Mouse IgG, light-chain specific, Jackson, 115-035-174, Sheep IgG HRP-conjugated Antibody, R&D SYSTEMS, HAF016). HRP signal was detected using ECL Western Blotting Detection Reagent (GE Healthcare) prior to development on autoradiography films (Amersham Hyperfilm) on an X-ray film processor (Xonograph Imaging Systems Compact XA). Membranes were acquired with EPSON software (version 3.9.2.1) and protein bands intensity was quantified with ImageJ and normalised on Ponceau staining. Uncropped scans of Western blots are provided in the Source Data file and at the end of the Supplementary Information file.

## Primary sympathetic neuronal cultures

The superior cervical ganglia (SCGs) were dissected from euthanized P0-P3 C57BL/6J mouse pups, cleaned from the surrounding tissues and incubated in 0.25% Trypsin-EDTA (ThermoFisher Scientific) for 30 min at 37 °C followed by 2 mg/ml Type II Collagenase (ThermoFisher Scientific) in sterile PBS for 30 min at 37 °C. After digestion, ganglion cells were mechanically dissociated using a sterile pipette, transferred into complete medium (DMEM with 10% heat-inactivated FBS, 100 U/ml penicillin and 100 μg/ml streptomycin−ThermoFisher) and plated at a very low density on 12 mm glass coverslips previously coated with 100 μg/ml poly-ᴅ-lysine (Sigma Aldrich) and 10 μg/ml Laminin (Scientific Laboratory Supplies). Neurons were grown for 24 h in complete medium supplemented with 1 or 10 ng/ml NGF (nerve growth factor, Alomone Labs) with or without the indicated doses of murine recombinant Negr1 (Sino Biological), before being fixed for immunofluorescence. For compartmentalised cultures, dissociated neurons were seeded on the left side (somatic chamber) of microfluidic devices (XC450 Xona Chips, Xona Microfluidics) previously coated with 500 μg/ml poly-ᴅ-Lysine and 10 μg/ml laminin. The complete medium in both compartments was supplemented with 10 ng/ml NGF, and cells were maintained at 37 °C and 5% CO₂. After 3−4 days in vitro, once the axons had successfully grown and reached the right compartment (axonal chamber), the growth medium in the axonal chamber was replaced with the conditioned media obtained from either wt or *ob/ob* fat explants (diluted 1:2 in neuronal complete medium without NGF, with or without 500 ng/ml murine recombinant Negr1) and a small volume of fresh growth medium was added to the somatic chamber to create the pressure gradient required for the fluidic isolation of the two compartments. After 24 h, brightfield images of the somatic compartment were randomly acquired and, in some cases, the whole microfluidic was fixed and processed for immunofluorescence.

## Immunofluorescence and quantification of axonal length, branching and neurodegeneration

At the end of the experiment, primary sympathetic neurons were fixed in 4% PFA in PBS for 15 min at RT (30 min at RT for neurons seeded on microfluidic devices), washed with PBS and permeabilised for 5 min in 0.3% Triton in PBS. After blocking with 3% donkey serum in PBS for 1 h at RT, neurons were incubated O.N. at 4 °C with anti-β3-Tubulin (abcam, ab18207, 1:100) diluted in blocking solution. After washing with PBS, cells were incubated with a fluorescent anti-rabbit antibody (Donkey anti-Rabbit IgG (H + L) Highly Cross-Adsorbed Secondary Antibody, Alexa Fluor 647, A-31573) in blocking medium for 1 h at RT, washed with PBS and counterstained with DAPI. Images were acquired on a Leica DM IL LED inverted microscope equipped with a DFC3000 G

camera and visualised with LAS X software (version 3.6.0.20104). For the quantification of the size and complexity of axon arborisations, 80 to 120 neurons from 3 independent experiments were analysed per each condition. The total axonal length was quantified with the NeuronJ plugin (ImageJ, Fiji, version 2.0.0), the total branches and principal neurites were counted manually. Statistical analysis was performed using ordinary one-way ANOVA followed by Tuckey's multiple comparison test (GraphPad, Prism, versions 8.01 and 9.2.0). To quantify axonal degeneration in compartmentalised neuronal cultures, swellings along neurites in the axonal chamber were counted in random 40x brightfield live images and normalised on the total length of axons contained in each picture assessed with NeuronJ. 10 to 20 random fields per microfluidics were analysed in 3 independent experiments, and statistical significance was calculated by unpaired, two-tailed independent Student's $t$-test (for 2 groups comparison) or ordinary one-way ANOVA followed by Tuckey's multiple comparison test (for 3 groups comparison) (GraphPad, Prism, versions 8.01 and 9.2.0).

### Statistical analysis on MS data

Data represent relative protein abundances normalised by the total amount of peptides, to consider the variation in abundances between samples, and scaled on the average of all samples. Both normalisation and scaling were performed on the total list of detected proteins, before filtering for secreted ones. Secreted proteins were selected according to prediction softwares SignalP (version 4.1)[22] and MatrisomeDB 2.0[23]. Proteins with a strong prediction index for mitochondrial localisation according to TargetP (version 2.0)[56] were excluded from further analysis. The datasets were further filtered to keep proteins with less than 20% missing values or with more than 90% in one of the examined groups and less than 10% for the rest of the groups. In the latter case missing values of the group, which presented more than 90% of missing values were imputed with zeros. All remaining missing values were imputed with KNN-Impute method using $k = 3$. The relative quantities of the proteins were scaled using log2 transformation. The limma package[57] was used to compare different groups using the Ebayes algorithm performing paired analysis when needed. The nominal $p$-values were adjusted for multiple testing using Benjamini–Hochberg method[58], and a threshold of 0.05 was used to infer statistically significant changes. Enrichment analysis was conducted with David Bioinformatics (version 6.8)[59], using proteins identified as secreted in each dataset as background lists in each analysis (526 hits for the first dataset, 415 hits for the second dataset). Principal component analysis[60] was conducted using the scikit-learn python library[61] version 0.19.2 and 3D scatterplots were visualised using python the matplotlib library (v. 3.5.1). Data visualisation was performed using the ggplot2 package (version 2.2.1) of R environment (version 4.1.2) and GraphPhad Prism (versions 8.01 and 9.2.0).

### Reporting summary

Further information on research design is available in the Nature Portfolio Reporting Summary linked to this article.

## Data availability

All data generated or analysed during this study are included in this published article and in its supplementary information files. Source Data are provided with this paper. The mass-spectrometry proteomics data generated and analysed during the current study have been deposited to the ProteomeXchange Consortium via the PRIDE partner repository with the dataset identifier PXD031271 and https://doi.org/10.6019/PXD031271 (https://doi.org/10.6019/PXD031271). The following servers/databases were used for data search/analysis: SignalP 4.1 (https://services.healthtech.dtu.dk/service.php?SignalP-4.1), TargetP 2.0 (https://services.healthtech.dtu.dk/service.php?TargetP-2.0), Matrisome DB 2.0 (http://matrisomeproject.mit.edu), UniProtKB/Swiss-Prot version May 2018 (https://ftp.uniprot.org/pub/databases/uniprot/previous_releases/release-2018_05/knowledgebase/uniprot_sprot-only2018_05.tar.gz). Source data are provided with this paper.

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

## Acknowledgements

We sincerely thank the Wohl Cellular Imaging Centre (King's College London) where the confocal microscopy images were acquired. We are also grateful to Dr. Matthew White for helping us setting up primary sympathetic neuronal cultures. M.M. is a British Heart Foundation (BHF) Chair Holder (CH/16/3/32406) with BHF programme grant support (RG/16/14/32397, RG/F/21/110053). M.M. received support from the BHF Centre for Vascular Regeneration with Edinburgh/Bristol

(RM/17/3/33381). M.M. is also supported by the Leducq Foundation ("PlaqOmics", 18CVD02) and VASCage-C (Research Centre on Vascular Ageing and Stroke), an R&D K-Centre of the Austrian Research Promotion Agency (COMET program—Competence Centres for Excellent Technologies) funded by the Austrian Ministry for Transport, Innovation and Technology, the Austrian Ministry for Digital and Economic Affairs and the federal states Tyrol, Salzburg and Vienna with the grant number FSG 868624. This study was also funded by the Deutsche Forschungsgemeinschaft (DFG, German Research Foundation)—Grant No. 236177352—SFB 1116, TPB10; GRK2576 vivid, P2 (M.G.).

## Author contributions

E.D. and M.M. conceived and designed the study; E.D., C.M.R. and J.B.B. performed experiments and analysed and interpreted data; U.M. performed most animal work; M.G. and A.B. performed HFD experiments and harvested the samples used in this paper; M.H. and K.T. carried out statistical analysis on all proteomics datasets; L.E.S. and S.A.B. performed LC-MS/MS analysis on all samples; A.V. and M.J. provided the human EAT samples; E.D. wrote the manuscript; M.M. reviewed the manuscript and secured the funding.

## Competing interests

The authors declare no competing interests.
