## [Peer Review File · Nature Communications]

Title: Reduced secretion of neuronal growth regulator 1 contributes to impaired adipose-neuronal crosstalk in obesityREVIEWER COMMENTS

Reviewer #1 (Remarks to the Author):

The manuscript by Duregotti et al. uses mass spectrometry to compare the secreted proteome of perivascular adipose tissue (PVAT) explants to more well studied white and brown fat depots. The authors note that the PVAT secretome is enriched for proteins involved in neuronal cell adhesion, consistent with the increase in sympathetic nerve projections in this depot. The authors then studied PVAT from ob/ob mice, which has decreased sympathetic projections. To explore underlying mechanisms, they used an in vitro microfluidic device and propose that neuronal growth regulator 1 (Negr1) has a neurotrophic function, as it may protect ob/ob neurites from degeneration. This work is potentially exciting, as it would add a new molecule to our still very incomplete understanding of factors involved in adipose-neuronal crosstalk. However, I have several serious concerns with the work here as detailed below. Without very extensive revision, the data are only preliminary and do not support the very broad conclusions made.

Major Points:

1. The methods used here have important flaws. By taking tissue explants, the authors by definition are transecting nerves and blood vessels. They then incubate these explants in serum free conditioned media (CM) for 24 hrs, where there is undoubtedly hypoxia and cell death. While the authors claim that the CM is devoid of intracellular proteins, only ~500 of the ~2400 proteins identified by mass spec are predicted to be secreted. Review of the mass spec data shows a large number of known intracellular proteins. How do the authors account for this? There needs to be some control for tissue viability. Moreover, in these heterogeneous explants, how can they make any statement about cell type of origin? In several places, they refer to Negr as adipocyte-derived, but this is never actually shown. They would need to show using mRNA and protein analysis from cultured adipocytes and/or fractionated adipose tissue that Negr1 is indeed adipocyte-derived. It seems much more likely that any Negr1 detected in their dataset comes from dead/dying nerve projections.

2. The authors use ob/ob mice as a model of obesity and repeatedly make broad claims about obesity-regulation. While ob/ob mice are extremely obese, they are deficient in leptin, which is not the case for the overwhelming majority of obese humans. These claims would be more compelling if they were also supported with data from a diet-induced mouse model. Moreover, leptin has recently been shown to play an important role in the sympathetic innervation of adipose tissues (Wang et al, 2020). The authors only mention this work in passing, but these data would serve as a nice control for benchmarking their later findings with Negr1.

3. The data on Negr1 presented in Figures 5 and 6 are very preliminary and the authors have certainly not presented findings that support the very broad claims made in the first paragraph of the discussion. First, the entire premise of neurodegeneration is made based on scoring the morphology of varicosities. Such an argument would only be convincing with a much deeper mechanistic exploration including

corroborating molecular markers. Second, the data in Figure 6 contains no explanation for how the dose of Negr1 was chosen or any controls to confirm that the Negr being used is in fact bioactive. Third, to truly confirm that Negr1 plays a role in this process gain and loss of function studies are needed and at least some of this needs to be done in vivo. The authors cite a paper describing a whole body Negr1 knockout mouse. To start, it would be interesting to know whether these mice have a defect in adipose tissue innervation.

Minor Points:

1. The authors present data on TH staining in epicardial fat from obese patients. This is potentially interesting, but is a distinct fat depot and really does not add to the work that is the focus of this manuscript.
2. The perilipin and Negr1 staining in Figure 6c did not work well. It is not possible from this image to conclude that Negr1 is on the adipocyte membrane and in the extracellular space.

Reviewer #2 (Remarks to the Author):

In the underlying manuscript the authors investigate different adipose tissue secretomes in ob/ob mice and humans using a proteomics-based approach. In particular, they compared secretory profiles of three distinct perivascular and three canonical depots. Epicardial fat derived from humans was used to validate the results. They found, that obesity is associated with significant changes of perivascular fat secretion pattern, which is accompanied by reduced sympathetic innervation. The adipokine Negr1 was identified as crucial mediator, whereby reduced secretion in obese state promotes the observed neurodegenerative effect.

As adipokines are key players in the communication between different organs, it is extremely valuable to characterize and dissect secretomes from distinct adipose tissue depots. Detailed information about modulation of secretion pattern in different metabolic states will help to understand the nature of obesity and therefore the pathophysiology of associated metabolic and cardiovascular disorders.

The entire proteomics workflow including sample preparation, mass spectrometry-based measurement and data analysis was carried out conscientiously. Moreover, identification and validation of Negr1 as crucial component in obesity mediated perivascular sympathetic neurodegeneration is convincing. In order to define potentially secreted proteins, the authors used filtering of the identified proteins. Only proteins were considered, which passed SignalP or Matrisome and were TargetP negative for mitochondrial localization. This very stringent strategy is straight forward but unfortunately may exclude proteins released by non-classical secretion processes or shuttled by extracellular vesicles especially exosomes. Due to the fact, that the non-classical secreted and exosomal proteins constitute an essential part of the adipokinome, this issue should be addressed at least in the discussion section. Better would

be to annotate non-classical secreted proteins utilizing bioinformatic tools like SecretomeP. For further investigations it is recommended also to address Adipokines shuttled by vesicles utilizing separated analysis subsequent to concentration of the supernatants.

Reviewer #3 (Remarks to the Author):

In this study Duregotti and colleagues performed proteomics analyses to define the secretory profiles of different adipose tissue (AT) depots from wt and obese (ob/ob) mice. This led to the observation that perivascular AT (PVAT) has a denser sympathetic innervation compared to other fat tissues. In ob/ob mice, they observed a loss of sympathetic axons in perivascular and SC depots. Mechanistically, this neurodegeneration linked to obesity was due, at least in part, to the downregulation of Negr1, a neurotrophic factor that is produced by adipocytes, as the addition of Negr1 partially restored the impaired neurotrophic effects of AT conditioned media from ob/ob mice in vitro. This is an interesting study shedding light on the adipose-neuronal crosstalk in obesity that thus may be suitable for publication on this journal after addressing a few comments as mentioned below.

- The authors used wt male C57BL/6J mice (Charles River) and ob/ob animals with the same genetic background (B6.Cg-Lepob 19 /J, Jackson Laboratories). These animals are not littermates and may differ more than expected. This should be mentioned as a limitation of the study.

- In Supplementary Figure 3a, the authors should display also the WB and relative analyses for BAT and AB PVAT.

- To improve the relevance of their findings in humans, it would be interesting to evaluate whether the denser sympathetic innervation of the PVAT in mice is observed also in human samples by comparing PVAT to VI WAT from the same patients. Is Negr1 detected in human samples? If so, is there an association between Negr1 and obesity?

- Was Negr1 detected in other AT depots other than PVAT? How do its levels vary among depots? And how do they change in ob/ob mice vs wt?

- I suggest to add a legend in panel a of Figure 6 to depict wt and ob/ob mice.

REVIEWERS COMMENTS

Reviewer #1 (Remarks to the Author):

The manuscript by Duregotti et al. uses mass spectrometry to compare the secreted proteome of perivascular adipose tissue (PVAT) explants to more well studied white and brown fat depots. The authors note that the PVAT secretome is enriched for proteins involved in neuronal cell adhesion, consistent with the increase in sympathetic nerve projections in this depot. The authors then studied PVAT from ob/ob mice, which has decreased sympathetic projections. To explore underlying mechanisms, they used an in vitro microfluidic device and propose that neuronal growth regulator 1 (Negr1) has a neurotrophic function, as it may protect ob/ob neurites from degeneration. This work is potentially exciting, as it would add a new molecule to our still very incomplete understanding of factors involved in adipose-neuronal crosstalk. However, I have several serious concerns with the work here as detailed below. Without very extensive revision, the data are only preliminary and do not support the very broad conclusions made.

Major Points:

1. The methods used here have important flaws. By taking tissue explants, the authors by definition are transecting nerves and blood vessels. They then incubate these explants in serum free conditioned media (CM) for 24 hrs, where there is undoubtedly hypoxia and cell death. While the authors claim that the CM is devoid of intracellular proteins, only ~500 of the ~2400 proteins identified by mass spec are predicted to be secreted. Review of the mass spec data shows a large number of known intracellular proteins. How do the authors account for this? There needs to be some control for tissue viability. Moreover, in these heterogeneous explants, how can they make any statement about cell type of origin? In several places, they refer to Negr as adipocyte-derived, but this is never actually shown. They would need to show using mRNA and protein analysis from cultured adipocytes and/or fractionated adipose tissue that Negr1 is indeed adipocyte-derived. It seems much more likely that any Negr1 detected in their dataset comes from dead/dying nerve projections.

We would like to thank the reviewer for this detailed comment. We acknowledge that an *ex vivo* incubation cannot recapitulate the *in vivo* environment. However, this same or similar procedures have already been used in the past in published articles to collect the conditioned media of adipose tissue explants^{1,2,3,4} and some of them were used as a reference to set up our protocol. We think that our protocol represents a good compromise to keep the tissue viable while obtaining reliable data on protein secretion as pointed out by the other reviewers. F12 supplement was used to compensate for the lack of FBS, which would have caused too much interference and contamination for the subsequent MS analysis on conditioned media. The freshly explanted fat chunks were finely minced and extensively washed to i) maximise the surface/volume ratio and favour the oxygen and nutrients diffusion across the tissue and ii) to minimise contamination from endogenous plasma proteins. The overall viability of tissues was confirmed by checking the pH of the conditioned medium at the beginning and the end of every experiment: in all cases, pH was found to be unchanged, indicating a low, negligible rate of cell death. The following sentence has been added at page 18, lanes 16-18 in the Methods section:

“The pH of the conditioned media was always found to be unchanged at the beginning and at the end of the incubation period, indicating a low, negligible rate of cell death in the tissue”.

We also acknowledge that a high number of proteins detected by MS in conditioned media samples are intracellular proteins. Intracellular proteins are abundant and there is always some leakage of cellular proteins either during the tissue preparation or the time course of the incubation. A low rate of cell death is expected even

in monolayer cell-cultures. In support of this, we observed that a similar percentage of intracellular proteins is detected in the conditioned media of cultured adipocytes in a recently published paper⁵. Here, differentiated human adipocytes are cultured in DMEM-F12 (without FBS) and the conditioned media is analysed by MS. Even if the incubation time is shorter (4 instead of 24 hours) the final percentage of secreted/total proteins detected is 25% (340 signal peptide-containing classically secreted plus 131 non-classically secreted, out of 1866 total detected) vs. our 22% (526 classically secreted on 2407 total). If non-classically secreted proteins (SecretomeP⁶-selected, 459 additional hits) were included in our list, this would bring the percentage of secreted proteins up to 40%. This demonstrates that the detection of intracellular proteins at a similar degree also occurs in monolayer-cultured cells, where the exposure to oxygen and nutrients is expected to be optimal.

As a further validation of the quality of our protocol, we provide evidence for enrichment of intracellular and secreted proteins in lysates and conditioned media respectively (**Fig. 1d** and tyrosine hydroxylase in **Fig. 3f** and **3g**, which are provided below in an uncropped version for the reviewer) and we also demonstrate that proteins of particular interest are detected in conditioned media following the release of soluble isoforms. This is verified for neuronal cell adhesion molecules, for which only the smallest isoforms are detected in secretomes (**Fig. 2f**) and for Negr1, whose detection in CM is almost completely abolished when AT explants are incubated in presence of GM6001, a broad-spectrum matrix metalloproteinases inhibitor (new **Fig. 6e** and **Supplementary Fig. 4c**, also shown here). However, we do agree with the reviewer that our secretomes are not “devoid” of intracellular proteins – as reported in the previous version of the paper -, so we have now amended the sentence at page 5, lines 6-8:

“Immunoblotting confirmed that secreted adipokines – such as adiponectin – were much more abundant in conditioned media compared to intracellular proteins that were enriched in AT lysates (Fig. 1d)”.

Figure 3f (lower panel) and 3g: uncropped version

New Figure 6e

New Supplementary Figure 4c

Figure 3f (lower panel) and 3g: uncropped version: this figure is shown here in an uncropped version to better show the enrichment of intracellular tyrosine hydroxylase in PVAT lysates compared to the corresponding conditioned media. **New Figure 6e** and **new Supplementary figure 4c:** western blot showing Negr1 in conditioned media of AR and TH PVAT and SC WAT explants incubated in absence or presence of broad-spectrum ECM metalloproteinase inhibitor GM6001. Abundance of secreted adipokines plasminogen activator inhibitor 1 and adiponectin are not affected by GM6001 treatment as they are not released by ECM MMP cleavage.

We also agree that the adipose parenchyma is composed by heterogeneous cell populations:

- For proteins detected in the secretomes, we refer to them as PVAT-derived.
- For proteins of interest that are further investigated, we do provide evidence of their cellular origin:

this is demonstrated for neuronal-derived cell-adhesion molecules Ncam1, L1cam and Ch11, which co-localise with tyrosine hydroxylase positive sympathetic neurites in perivascular sections (**Fig. 2d**) and are also expressed by primary murine sympathetic neurons *in vitro* (**Supp. Fig. 4b**). For Negr1, in addition to the immunofluorescence shown in the new version of **Fig. 6c** (also shown here), we now also provide a Western blot showing that after digestion of AT explants, Negr1 is only detectable in floating adipocytes and not in the pelleted stromal vascular fraction (SVF, **Fig. 6d**, also shown here). The same Western blot also further excludes a neuronal origin of Negr1, since sympathetic neurites distribute across the two fractions (as they are non-nucleated axonal fragments), but Negr1 is completely undetectable in the SVF. This is in addition to **Supp. Fig. 4b**, which clearly

shows the lack of Negr1 expression by primary cultured sympathetic neurons. The following sentences have been added to the manuscript (page 10, lines 22-25 and page 11, line 1):

“This is further confirmed by digestion and fractionation of AT explants (Fig. 6d). Negr1 is only detected in isolated floating adipocytes and not in the stromal vascular fraction (SVF). Negr1 protein is neither expressed by primary sympathetic neurons (Supplementary Fig. 4b) nor by sympathetic axonal fragments in AT tissues, that are present in both fractions after digestion due to the lack of nuclei (Fig. 6d)”.

Furthermore, a new section *“AT explants digestion and fractionation”* has been added to the Methods section, page 18, line 1 and page 19, lines 1-11.

New Figure 6c

New fig.6d

New Figure 6c: this figure is intended to replace previous fig. 6c and better show the expression of Negr1 by perilipin-positive adipocytes in PVAT sections. **New Figure 6d:** western blot on SVF and isolated adipocytes obtained from ex vivo digestion of VI, SC and BAT depots. While tyrosine hydroxylase (expressed by sympathetic fragments) redistribute across the two fractions, Negr1 is only detected in perilipin-expressing isolated adipocytes.

2. The authors use ob/ob mice as a model of obesity and repeatedly make broad claims about obesity-regulation. While ob/ob mice are extremely obese, they are deficient in leptin, which is not the case for the overwhelming majority of obese humans. These claims would be more compelling if they were also supported with data from a diet-induced mouse model. Moreover, leptin has recently been shown to play an important role in the sympathetic innervation of adipose tissues (Wang et al, 2020). The authors only mention this work in passing, but these data would serve as a nice control for benchmarking their later findings with Negr1.

We thank the reviewer for this valuable comment. Indeed, the recent work by Wang et al. (Nature 2020) demonstrated that leptin plays an important role in the sympathetic innervation of AT and our findings on Negr1 extend on this seminal observation. The paper is now further discussed in the discussion, page 14, lines 7-8:

“Of note, a similar observation was also made in SC WAT and in BAT, confirming a recent report by Wang and colleagues⁷.”

and page 14, lines 11-15:

“While Wang and colleagues elegantly demonstrated a central nervous system-mediated mechanism that links systemic leptin resistance in obesity to SC WAT and BAT denervation in mice⁷, additional regulatory processes may also modulate innervation at a local level⁸. Indeed, fat secreted factors have previously been implicated in the paracrine regulation of intra-adipose sympathetic plasticity in the context of cold-induced beiging^{9,10}.”

As requested, we now provide data of a diet-induced obese (DIO) mouse model. Mice were fed a 60% high fat diet for 16 weeks, hence the delay in our re-submission. We include additional experiments assessing the sympathetic innervation and Negr1 expression in the AR PVAT, TH PVAT and SC WAT (which are the main depots we analysed in *ob/ob* mice) of DIO mice, compared with age-matched low-fat diet-fed animals. Negr1 expression in tissue lysates is a good index of its secretion levels, as proven by our experiments showing that i) Negr1 expression is reduced not only in secretomes, but also in AR and TH PVAT and SC WAT lysates from *ob/ob* mice (new Fig. 6f and 6g and Supp. Fig. 4f, also shown below) and ii) soluble Negr1 is cleaved by ECM MMP from the surface of adipocytes (see new Fig. 6e and Supp. Fig. 4c also shown here at page 2). In DIO mice we do see a significant decrease of both tyrosine hydroxylase (Supp. Fig. 3d, also shown here, next page) and Negr1 (Supp. Fig. 4g, also shown here, next page). Only in two instances, the decrease falls short of statistical significance: for tyrosine hydroxylase in AR samples ($p=0.064$) and for Negr1 in TH samples ($p=0.188$). This is likely ascribable to the higher heterogeneity we see across samples as the DIO mouse represents a milder model of obesity in which some pathological aspects of the disease are expected to be less pronounced than in the *ob/ob* mouse. However, a reduction in sympathetic innervation and Negr1 expression is observed in most analysed fat depots of DIO animals, further supporting our observations in *ob/ob* mice. The following sentences have been added to the updated version of the manuscript: page 9, lines 4-6:

“Notably, tyrosine hydroxylase was also decreased in the AR and TH PVAT and in the SC WAT of diet-induced obese (DIO) mice (Supplementary Fig. 3d), indicating that a loss of adipose sympathetic innervation also occurs in another model of murine obesity”.

page 11, lines 10-12:

*“A significant decrease of Negr1 is also detected in AR PVAT and SC WAT lysates of DIO mice compared to lean controls (Supplementary Fig. 4g), while a higher variability was observed in the TH PVAT of DIO mice compared to *ob/ob* mice”.*

New Fig. 6f, 6g and new Supplementary Figure 4f: Western blot and corresponding quantifications showing that Negr1 expression is significantly reduced in the TH (new Fig. 6f) and AR PVAT (new fig. 6g) and in the SC WAT lysates (new Supplementary Fig. 4f) of *ob/ob* mice.

Supplementary Figures 3d and 4g: Western blot showing TYR.H. (new Supplementary Fig. 3d) and Negr1 (new Supplementary Fig. 4g) expression in TH and AR PVAT and in SC WAT lysates of wt mice fed with a HFD or a matched low-fat control diet (CD) for 16 weeks from 5 weeks of age.

3. The data on Negr1 presented in Figures 5 and 6 are very preliminary and the authors have certainly not presented findings that support the very broad claims made in the first paragraph of the discussion. First, the entire premise of neurodegeneration is made based on scoring the morphology of varicosities. Such an argument would only be convincing with a much deeper mechanistic exploration including corroborating molecular markers. Second, the data in Figure 6 contains no explanation for how the dose of Negr1 was chosen or any controls to confirm that the Negr being used is in fact bioactive. Third, to truly confirm that Negr1 plays a role in this process gain and loss of function studies are needed and at least some of this needs to be done in vivo. The authors cite a paper describing a whole body Negr1 knockout mouse. To start, it would be interesting to know whether these mice have a defect in adipose tissue innervation.

Concerning the first point raised by the reviewer, we would like to point out that the decreased sympathetic innervation observed in *ob/ob* animals is demonstrated by a significant reduction of tyrosine hydroxylase protein levels in extracts obtained from PVATs, SC WAT and BAT (as shown in Fig. 4a and 4b and in Supp. Fig. 3a, b and c). Western blotting is a more reliable quantitative method than immunofluorescence staining and provides overall protein abundance in the whole tissue. Also, it is not biased by selection of areas of interest or differences in fluorescence intensity that might derive from artifacts or different background noise. Our Western blot data are further corroborated by IF on AT sections, where we analysed the morphology rather than the fluorescence intensity of neurites. The increased fragmentation detected in *ob/ob* animals is in line with a progressive degeneration of sympathetic projections^{11,12} – labelled with tyrosine hydroxylase, the only molecular marker specific for sympathetic neurons (Fig. 4c, d and e).

Regarding the second point raised, we now provide more clarity on how we choose the dose on Negr1 used in our *in vitro* experiments. We have added a new supplementary figure (Supp. Fig. 4h, also shown here) reporting a pilot experiment that was initially performed testing Negr1 doses ranging from 1 to 1000 ng/ml on primary murine sympathetic neurons. This experiment demonstrated that the lowest dose found to be significantly effective in increasing axonal elongation was 500 ng/ml. This same dose was then found to also promote axonal branching (see Fig. 7b and 7c). This can now be found at page 11, lines 18-25 and page 12, lines 1-2:

*“To investigate whether soluble Negr1 is indeed endowed with neurotrophic functions, we performed a dose-finding experiment culturing primary sympathetic neurons at very low density for 24 hours in presence of 1 ng/ml NGF (which is essential for neuronal growth and survival) and increasing concentrations of soluble recombinant Negr1 (0, 1, 10, 100, 500, 1000 ng/ml). As shown in **Supplementary Figure 4h**, Negr1 addition resulted in an enhanced axonal elongation, with the lowest dose found to be significantly effective being 500 ng/ml. Additional experiments showed that the same dose of Negr1 significantly increased the total axonal length and the number of branching points in presence of both 1 and 10 ng/ml NGF, while it had no effect on the average number of principal neurites emerging from the neuronal somata (**Fig. 7b and 7c**). This indicates that Negr1 significantly enhances the growth and arborization of distal sympathetic neurites”.*

This is also in line with a recently published paper⁹, where authors found adipocyte-secreted NRG4 to stimulate axonal branching and elongation when used at 100-1000 ng/ml *in vitro*. Moreover, as adipocyte-expressed Negr1 and sympathetic neurites are found to be in close juxtaposition in AT sections (see **Fig. 7a**), a rather high concentration of Negr1 is likely to be present in the microenvironment between the Negr1-expressing adipocyte membrane and the nearby axon. In our opinion, the fact that the used recombinant Negr1 is bioactive is proven by its enhancing effect on axonal arborization.

For the reviewer’s information, we also include here an experiment showing that addition of recombinant Negr1 induces the phosphorylation of ERK in NGF-deprived neurons in a way similar to NGF, confirming that it is perceived by sympathetic neurons where it activates extracellular signal-regulated pathways.

New Supplementary Fig. 4h

Not shown in the manuscript

New Supplementary Fig. 4h: bar chart showing the quantification of the total axonal length measured on primary sympathetic neurons incubated with 1 ng/ml NGF and the reported concentrations of recombinant Negr1 for 24 hours. **Figure not shown in the manuscript:** for the reviewer’s knowledge, we present here a representative Western blot and the quantification of 3 independent experiments showing that addition of recombinant Negr1 induces the phosphorylation of ERK in primary murine sympathetic neurons. In more detail, neurons were deprived from NGF for 7 hours and then either NGF, Negr1 or a combination of the 2 factors were added to the cells for 10 minutes. Neurons were then lysed and processed by Western blot.

Concerning the third point raised, Negr1 knock-out mice are not commercially available, so we decided to attempt an *in vivo* rescue experiment instead. This was performed by implanting subcutaneous osmotic minipumps connected to a catheter delivering recombinant Negr1 in the subcutaneous WAT in *ob/ob* mice (new **Fig. 7f**). SC WAT was chosen as a target because of the poor accessibility of the peri-aortic PVAT. However, decreased sympathetic innervation and Negr1 secretion/expression were also observed in the SC WAT of obese animals, making this easy-accessible depot a good substitute for a proof of principle experiment (see **Supp. Fig. 3a, 4e and 4f**). As shown in **Fig. 7g and 7h**, the local delivery of Negr1 in the SC WAT for 2 weeks did not induce any significant change in the mice weight, nor in the weight of the SC WAT depot itself. However, a significant increase of tyrosine hydroxylase was observed in the SC WAT protein extracts of *ob/ob* mice administered with Negr1 (new **Fig. 7i**). This suggests that Negr1 does promote the re-innervation of SC WAT on obese animals. These data are now shown in **Fig. 7** and incorporated in the manuscript, page 12, lines 5-13:

*“Finally, we decided to test the effect of Negr1 on adipose sympathetic innervation in vivo. Due to the poor accessibility of PVAT, we decided to focus on the SC WAT, which in is also characterised by decreased innervation and Negr1 secretion in ob/ob mice (see **Supplementary Fig. 3a** and **4e**). Subcutaneous osmotic minipumps connected to a catheter delivering recombinant Negr1 (1 µg/day) were implanted at the level of the inguinal SC WAT of ob/ob mice (see **Fig. 7f**) for 2 weeks. As shown in **Fig. 7g** and **7h**, the local delivery of Negr1 did not induce any significant change in the mice weight, nor in the weight of the SC WAT depot. However, an increase of tyrosine hydroxylase was observed in the SC WAT lysates of ob/ob mice upon administration of Negr1 (**Fig. 7i**), demonstrating that Negr1 promotes the sympathetic axonal growth *iv vivo*”.*

A new section “*In vivo osmotic minipumps implantation*” has also been added to the Methods section (page 17, lines 6-19).

New Fig. 7f

New Fig. 7g

New Fig. 7h

New Figure 7f: cartoon showing how the osmotic minipump was subcutaneously implanted in *ob/ob* mice and the connected catheter's end positioned within the SC WAT. **New Figures 7g and 7h:** local Negr1 delivery for 2 weeks via catheter-connected osmotic minipump did not affect the mice's nor the SC WAT weight. **New Figure 7i:** Western blot and quantification showing that tyrosine hydroxylase is increased in SC WAT lysates of mice receiving recombinant Negr1 with respect to mice receiving saline only (vehicle).

New Fig. 7i

Minor points:

1. The authors present data on TH staining in epicardial fat from obese patients. This is potentially interesting, but is a distinct fat depot and really does not add to the work that is the focus of this manuscript.

The epicardial fat was collected from an area very close to the right coronary artery, thus making it a suitable example of perivascular fat. Moreover, a reduced sympathetic innervation has been described for other AT depots in obese mice^{7,13}, like SC AT and BAT (which is now also confirmed in our paper, see **Supp. Fig. 3a** and **3c**). This suggests that not only the perivascular fat is affected but most kind of AT depots in obese mice. We believe it is a strength of our manuscript to also include data on human AT.

2. The perilipin and Negr1 staining in Figure 6c did not work well. It is not possible from this image to conclude that Negr1 is on the adipocyte membrane and in the extracellular space.

We appreciate the reviewer's concern. We have now added enlarged crops which better show the localization of Negr1 (Fig. 6c, also shown here at page 3). We have also included a negative control of sections only incubated with the fluorescent secondary antibodies, which shows the absence of specific staining – although some autofluorescence is produced by lipid droplets contained in the tissue. Of note, we do not expect all Negr1 signal to be localised on the plasma membrane, as following the permeabilization of the sections also the intracellular one – such as that being targeted to the membrane – is recognised and bound by the antibody.

The expression of Negr1 by adipocytes is now also proven by Western blot on isolated primary adipocytes and SVF (new Fig. 6d also shown here at page 3). Its localisation on the plasma membrane is further supported by the reduced abundance in the conditioned media following incubation of explants in presence of MMP wide-spectrum inhibitor GM6001 (new Fig. 6e and new Supp. Fig. 4c also shown here at page 2).

Reviewer #2 (Remarks to the Author):

In the underlying manuscript the authors investigate different adipose tissue secretomes in ob/ob mice and humans using a proteomics-based approach. In particular, they compared secretory profiles of three distinct perivascular and three canonical depots. Epicardial fat derived from humans was used to validate the results. They found, that obesity is associated with significant changes of perivascular fat secretion pattern, which is accompanied by reduced sympathetic innervation. The adipokine Negr1 was identified as crucial mediator, whereby reduced secretion in obese state promotes the observed neurodegenerative effect.

As adipokines are key players in the communication between different organs, it is extremely valuable to characterize and dissect secretomes from distinct adipose tissue depots. Detailed information about modulation of secretion pattern in different metabolic states will help to understand the nature of obesity and therefore the pathophysiology of associated metabolic and cardiovascular disorders.

The entire proteomics workflow including sample preparation, mass spectrometry-based measurement and data analysis was carried out conscientiously. Moreover, identification and validation of Negr1 as crucial component in obesity mediated perivascular sympathetic neurodegeneration is convincing. In order to define potentially secreted proteins, the authors used filtering of the identified proteins. Only proteins were considered, which passed SignalP or Matrisome and were TargetP negative for mitochondrial localization. This very stringent strategy is straight forward but unfortunately may exclude proteins released by non-classical secretion processes or shuttled by extracellular vesicles especially exosomes. Due to the fact, that the non-classical secreted and exosomal proteins constitute an essential part of the adipokinome, this issue should be addressed at least in the discussion section. Better would be to annotate non-classical secreted proteins utilizing bioinformatic tools like SecretomeP. For further investigations it is recommended also to address Adipokines shuttled by vesicles utilizing separated analysis subsequent to concentration of the supernatants.

We are very grateful to the reviewer for his approval of our extensive proteomics efforts, and for the positive comments on the technical workflow and results analysis. We appreciate that some secreted proteins may have been excluded from our stringent analysis, and indeed further filtering our 2407 list (Supplementary Table 1) with SecretomeP led to the identification of additional 459 non-classically secreted proteins, raising the percentage of secreted proteins from 22% to 41% among all proteins detected. This is further increased to 45% when including proteins reported as “extracellular/secreted” by Gene Ontology.

In this study, we decided to be as stringent as possible in terms of secreted protein selection in order to minimise “false positives”. However, for future works we will certainly consider including non-classically secreted proteins as this could bring in other important – and probably still undiscovered - adipokines.

We are also planning to isolate extracellular vesicles in our secretomes and profile their protein content, as we also think this would add to the overall knowledge about AT as an endocrine organ.

These limitations are now discussed in the discussion, page 13, lines 5-10:

“Proteomics analysis on conditioned media of 3 perivascular and 3 canonical fat depots (VI and SC WAT, BAT) led to the identification of 526 secreted proteins. These secreted proteins were either endowed with a signal peptide for secretion (SignalP¹⁴) or listed in the Matrixome¹⁵ database. While our analysis achieved a much greater proteome coverage than the previous publications to date^{16,17}, another source of secreted proteins are extracellular vesicles and exosomes, and additional studies will be required to specifically profile their protein content.”

Reviewer #3 (Remarks to the Author):

In this study Duregotti and colleagues performed proteomics analyses to define the secretory profiles of different adipose tissue (AT) depots from wt and obese (ob/ob) mice. This led to the observation that perivascular AT (PVAT) has a denser sympathetic innervation compared to other fat tissues. In ob/ob mice, they observed a loss of sympathetic axons in perivascular and SC depots. Mechanistically, this neurodegeneration linked to obesity was due, at least in part, to the downregulation of Negr1, a neurotrophic factor that is produced by adipocytes, as the addition of Negr1 partially restored the impaired neurotrophic effects of AT conditioned media from ob/ob mice in vitro. This is an interesting study shedding light on the adipose-neuronal crosstalk in obesity that thus may be suitable for publication on this journal after addressing a few comments as mentioned below.

- The authors used wt male C57BL/6J mice (Charles River) and ob/ob animals with the same genetic background (B6.Cg-Lepob 19 /J, Jackson Laboratories). These animals are not littermates and may differ more than expected. This should be mentioned as a limitation of the study.

We highly appreciate the reviewer’s positive feedback on our work, and we thank the reviewer for this comment as it allowed us to spot a mistake in the Methods section, which has now been amended. Both wt and ob/ob age-matched mice were purchased together at the same time from the same Charles River facility, and this should minimise variability. We do agree that working with littermates is the more reliable option, but in many published articles this is not specified for wt and ob/ob mice. Moreover, as ob/ob and wt mice have the same C57BL/6J background, we expect that the genetic variability among them is kept to a minimum by the frequent backcrosses performed at certified animal facilities. However, we now include this point in the Methods section, at page 16, line 14-16:

“Although wt and ob/ob age-matched mice employed here were not littermates, this is in line with many published articles^{7,18} and we expect this not to represent a significant limitation for this study.”

We also would like to point out that the findings in wt and ob/ob age-matched mice were now replicated in independent experiments using a diet-induced obese (DIO) mouse model (see comments to reviewer 1).

- In Supplementary Figure 3a, the authors should display also the WB and relative analyses for BAT and AB PVAT.

Western blots and quantifications of tyrosine hydroxylase in BAT and AB PVAT have now been added to Supplementary fig. 3 (new **Supp. Fig. 3b** and **3c**, also shown below). However, as specified in the manuscript, the massive decrease of tyrosine hydroxylase detected in *ob/ob* AB PVAT samples is most likely the consequence of a dissection-related artifact (see page 8, lines 12-16 and **Supp. Fig. 2c**). The manuscript has been amended accordingly (page 9, lines 1-4):

“A significant reduction of tyrosine hydroxylase was also detected in SC fat lysates from ob/ob mice, accompanied by lower levels of UCP1 (Supplementary Fig. 3a), as well as in lysates from AB PVAT (Supplementary Fig. 3b) and from BAT (Supplementary Fig. 3c).”

New Supplementary Fig. 3b

New Supplementary Figures 3b and 3c: Western blot and corresponding quantifications showing that TYR.H. is significantly reduced in AB PVAT and BAT of *ob/ob* mice compared to wt.

New Supplementary Fig. 3c

- To improve the relevance of their findings in humans, it would be interesting to evaluate whether the denser sympathetic innervation of the PVAT in mice is observed also in human samples by comparing PVAT to VI WAT from the same patients. Is Negr1 detected in human samples? If so, is there an association between Negr1 and obesity?

We thank the reviewer for raising this point. We only obtained epicardial AT (EAT) samples from the patients involved in this study, so we are not able to compare the EAT sympathetic innervation with that of other AT depots. We will have to address this in future. Currently, we cannot obtain human AT due to COVID-19.

We did stain Negr1 in the same EAT lysates shown in fig. 4g, and the results are shown here for the reviewer's information. Out of the 20 samples analysed, 2 were identified as outliers for Negr1 (but not for tyrosine hydroxylase, red-framed in Western blot) according to the Iglewicz and Hoaglin's robust test for multiple outliers (two-sided test, modified Z-score ≥ 3.5). While no significant Negr1/BMI correlation is observed when considering all the 20 samples (continuous line in chart and r and p-value within continuous frame), the negative correlation does become significant after excluding those 2 outliers (dashed line in chart and r and p-value in red within dashed frame; outliers shown as bright red spots). A similar outcome is observed when running a tyrosine hydroxylase to Negr1 correlation analysis, where a much higher significance is reached after removing those 2 outliers (see attached chart). While the highly positive and significant correlation in the remaining 18 samples supports the existence of a functional relationship between Negr1 and sympathetic innervation extent, we are currently unsure about what accounts for the very high Negr1 levels in our two outliers. If the reviewer agrees, we would prefer not to include these data in the current version of the manuscript, even though removing the two outliers would not affect the significance of all other analysis performed (see **Fig. 4 g,h,i** and **j**). More work needs to be done in the

context of human obesity to understand the mechanisms responsible for the higher inter-individual variability for Negr1 compared to tyrosine hydroxylase. In addition, Negr1 down-regulation might not be the only factor affecting the adipose innervation (see⁷).

Not shown in the manuscript

Western blot showing tyrosine hydroxylase (also in Fig. 4g) and Negr1 abundancies in the 20 EAT samples obtained from human subjects and employed in our study. Two samples were identified as outliers for Negr1 (red-framed in Western blot, bright red dots in charts) but not for tyrosine hydroxylase. In correlation charts, continuous lines and values within continuous frames refer to analysis performed on all the 20 samples, whereas dashed lines and values in red within dashed frames refer to analysis performed after excluding the 2 outliers. Spearman correlation analysis was performed.

- Was Negr1 detected in other AT depots other than PVAT? How do its levels vary among depots? And how do they change in ob/ob mice vs wt?

Negr1 was detected by untargeted MS in the conditioned media of all the 6 AT depots in our study. In addition to Fig. 6a showing Negr1 relative abundancies in AR and TH PVAT secretomes from wt and ob/ob mice, an additional figure (Supp. Fig. 4d) has now been added to display Negr1 levels in the conditioned media of other AT depots from the same mice: a significant decrease in ob/ob samples is observed in all the depots analysed. Negr1 is also detected in AT lysates (as shown in new Fig. 6f and 6g and new Supp. Fig. 4f and here at page 4), where its expression levels reflect the trend observed in corresponding conditioned media. The following sentences have been added to the manuscript, page 11, lines 6-10:

“The decreased abundance of Negr1 in ob/ob adipose secretomes extends to all AT depots (Supplementary Fig. 4d) and was validated for SC WAT (Supplementary Fig. 4e). Negr1 is significantly downregulated in AR and TH PVAT lysates as well as SC WAT lysates from ob/ob mice compared to wt (Fig. 6f and 6g and Supplementary Fig. 4f)”

Negr1 was also detected in the proteomics analysis performed in Fig. 1e, and its abundance across depots is shown here for the reviewer’s information.

New supplementary figure 4d: MS detected relative abundances of Negr1 in the conditioned media of AB PVAT, VI and SC WAT and BAT from wt and *ob/ob* mice. **Figure not shown in manuscript:** MS detected relative abundances of Negr1 in the conditioned media of AT samples from wt animals shown in Fig. 1e.

- I suggest to add a legend in panel a of Figure 6 to depict wt and *ob/ob* mice.

We thank the reviewer for spotting this error, a legend has now been added to **Fig. 6a**.

Data availability:

The mass-spectrometry proteomics data generated and analysed during the current study have been deposited to the ProteomeXchange Consortium via the PRIDE partner repository with the dataset identifier PXD031271 and 10.6019/PXD031271.

Username: reviewer_pxd031271@ebi.ac.uk

Password: PUNKyvmk

References for rebuttal:

1. Hocking, S. L., Wu, L. E., Guilhaus, M., Chisholm, D. J. & James, D. E. Intrinsic depot-specific differences in the secretome of adipose tissue, preadipocytes, and adipose tissue-derived microvascular endothelial cells. *Diabetes* **59**, 3008–3016 (2010).
2. Bailey-Downs, L. C. *et al.* Aging exacerbates obesity-induced oxidative stress and inflammation in perivascular adipose tissue in mice: a paracrine mechanism contributing to vascular redox dysregulation and inflammation. *J. Gerontol. A. Biol. Sci. Med. Sci.* **68**, 780–792 (2013).
3. Ruan, C. C. *et al.* Perivascular adipose tissue-derived complement 3 is required for adventitial fibroblast functions and adventitial remodeling in deoxycorticosterone acetate-salt hypertensive rats. *Arterioscler. Thromb. Vasc. Biol.* **30**, 2568–2574 (2010).
4. Roca-rivada, A. *et al.* Secretome analysis of rat adipose tissues shows location-specific roles for each depot type. *J. Proteomics* **74**, 1068–1079 (2011).
5. Deshmukh, A. S. *et al.* Proteomics-Based Comparative Mapping of the Secretomes of Human Brown and White Adipocytes Reveals EPDR1 as a Novel Adipokine. *Cell Metab.* **30**, 963-975.e7 (2019).
6. Bendtsen, J. D., Jensen, L. J., Blom, N., Von Heijne, G. & Brunak, S. Feature-based prediction of non-classical and leaderless protein secretion. *Protein Eng. Des. Sel.* **17**, 349–356 (2004).
7. Wang, P. *et al.* A leptin–BDNF pathway regulating sympathetic innervation of adipose tissue. *Nature* **583**, 1–6 (2020).
8. Zeng, X. *et al.* Innervation of thermogenic adipose tissue via a calyntenin 3 β –S100b axis. *Nature* **569**, 229–235 (2019).
9. Pellegrinelli, V. *et al.* Adipocyte-secreted BMP8b mediates adrenergic-induced remodeling of the neurovascular network in adipose tissue. *Nat. Commun.* **9**, 1–18 (2018).
10. Cao, Y., Wang, H. & Zeng, W. Whole-tissue 3D imaging reveals intra-adipose sympathetic plasticity regulated by NGF-TrkA signal in cold-induced beiging. *Protein Cell* **9**, 527–539 (2018).
11. Singh, K. K. *et al.* Developmental axon pruning mediated by BDNF-p75NTR-dependent axon degeneration. *Nat. Neurosci.* **11**, 649–658 (2008).
12. Park, K. J., Grosso, C. A., Aubert, I., Kaplan, D. R. & Miller, F. D. P75NTR-dependent, myelin-mediated axonal degeneration regulates neural connectivity in the adult brain. *Nat. Neurosci.* **13**, 559–566 (2010).
13. Jiang, H., Ding, X., Cao, Y., Wang, H. & Zeng, W. Dense Intra-adipose Sympathetic Arborizations Are Essential for Cold-Induced Beiging of Mouse White Adipose Tissue. *Cell Metab.* **26**, 686-692.e3 (2017).
14. Almagro Armenteros, J. J. *et al.* SignalP 5.0 improves signal peptide predictions using deep neural networks. *Nat. Biotechnol.* **37**, 420–423 (2019).
15. Naba, A. *et al.* The extracellular matrix: Tools and insights for the ‘omics’ era. *Matrix Biol.* (2016). doi:10.1016/j.matbio.2015.06.003
16. Alvarez-Llamas, G. *et al.* Characterization of the human visceral adipose tissue secretome. *Mol. Cell. Proteomics* **6**, 589–600 (2007).
17. Roca-Rivada, A. *et al.* Secretome analysis of rat adipose tissues shows location-specific roles for each depot type. *J. Proteomics* **74**, 1068–1079 (2011).
18. Pirzgalaska, R. M. *et al.* Sympathetic neuron-associated macrophages contribute to obesity by importing and metabolizing norepinephrine. *Nat. Med.* **23**, 1309–1318 (2017).

REVIEWER COMMENTS

Reviewer #1 (Remarks to the Author):

I have carefully reviewed the authors' rebuttal letter and revised manuscript, but continue to have significant concerns in the following areas:

1) Quality of the secretome data

While I appreciate the additional analyses the authors have done, the majority of proteins identified still appear to be intracellular proteins. Tissue explants will be particularly prone to cell death and increased elaboration of inflammatory markers once separated from their vascular supply and innervation. The authors say they have looked at pH of the media, but the actual data are not provided. Moreover, since cell culture media and incubation conditions provide pH buffering, it isn't clear to me that pH will necessarily change when cells in the explant are progressively dying. I would be more convinced about the degree of viability and comparability across samples if the authors had measured markers of cell death, stress, and inflammation. As a related matter, for the comparisons between depots and models, it isn't clear how the authors are normalizing data. Do they analyze an equivalent mass of tissue? This should be discussed and justified in detail.

2) Cellular origin of Negr1

While the authors have now provided adipose tissue fractionation data, this does not necessarily mean Negr1 is exclusively produced by adipocytes (or even produced by adipocytes at all). The authors should make this point more convincingly by either (a) differentiating SVF into adipocytes and showing that Negr1 mRNA is enriched in adipocytes and that Negr1 protein is secreted into the conditioned media by these adipocytes or (b) looking at Negr1 mRNA levels in published scRNA Seq and scNuc Seq datasets from adipose tissue. From the data provided, I am not at all convinced that Negr1 comes from adipocytes as opposed to from nerves innervating adipose tissue and associated vasculature. Finally, I am quite concerned that the imaging for Negr1 in Figure 7a entirely represents background signal. It looks like the tissue is diffusely stained, and the authors need to do a control with tissue that doesn't express Negr1 or secondary antibody alone to make it convincing that this is real signal.

3) Negr1 function

The authors have presented new experiments towards a function for Negr1, but a number of important issues remain, and the conclusions are vastly overstated. First, in Figure 7e, they claim that adding Negr1 to CM from ob/ob explants prevents neurodegeneration. The claim here and elsewhere regarding neurodegeneration is entirely based on subtle and non-quantitative morphological alterations. Molecular markers of neurodegeneration need to be analyzed. Second, if Negr1 is a neurotrophic factor, how do the authors explain their data in Figure S4D showing highest levels in depots that are least innervated? Third, the new in vivo experiment in Figure 7f-i needs much more corroboration. Specifically, what levels of Negr1 were achieved and was Negr1 only elevated locally or also systemically? Also, the TH imaging needs to be supported by tissue imaging for nerve fibers, as the authors have done elsewhere in the manuscript. The authors overall make excessive claims based on TH

westerns. This is a concern because the TH antibody used here has been shown to detect similar sized bands in both control and TH KO samples and thus may not be reliable (Fischer et al, Nat Med 2017, Fig S1).

Reviewer #2 (Remarks to the Author):

In the revised Nature Communications manuscript NCOMMS-21-25239A the authors addressed all concerns I made in the review sufficiently.

Reviewer #3 (Remarks to the Author):

In this study Duregotti and colleagues performed proteomics analyses to define the secretory profiles of different adipose tissue (AT) depots from wt and obese (ob/ob) mice. This led to the observation that perivascular AT (PVAT) has a denser sympathetic innervation compared to other fat tissues. In ob/ob mice, they observed a loss of sympathetic axons in perivascular and SC depots. Mechanistically, this neurodegeneration linked to obesity was due, at least in part, to the downregulation of Negr1, a neurotrophic factor that is produced by adipocytes, as the addition of Negr1 partially restored the impaired neurotrophic effects of AT conditioned media from ob/ob mice in vitro. This is a revised version of the original manuscript in which the authors addressed all my previous comments; hence, I now find the paper suitable for publication.

Reply to reviewer #1:

We thank Reviewer #1 for carefully reading our manuscript and have addressed the remaining concerns. Of course, any of the reviewer-only figures in the rebuttal could also be included in the Supplement if requested.

1) Quality of the secretome data. While I appreciate the additional analyses the authors have done, the majority of proteins identified still appear to be intracellular proteins. Tissue explants will be particularly prone to cell death and increased elaboration of inflammatory markers once separated from their vascular supply and innervation. The authors say they have looked at pH of the media, but the actual data are not provided. Moreover, since cell culture media and incubation conditions provide pH buffering, it isn't clear to me that pH will necessarily change when cells in the explant are progressively dying. I would be more convinced about the degree of viability and comparability across samples if the authors had measured markers of cell death, stress, and inflammation.

To further address the reviewer's comment, we have performed the following analysis:

- 1) As mentioned before, mass spectrometry is very sensitive and the detection of leaked intracellular proteins in conditioned media is unavoidable. Our results using tissue explants are comparable to previous proteomics results by others using conditioned media of cell cultures (Ali Khan et al., 2018)(Deshmukh et al., 2019). The percentage of secreted proteins we identified in adipose tissue explants (22%) is identical to that reported in adipocytes exposed to the same culture conditions (25%) and adopting the same workflow for proteomics data analysis (Deshmukh et al., 2019, published in *Cell Metabolism*). In fact, it would be even higher (40%) if non-classically secreted proteins (SecretomeP-identified) were included in our analysis as done in (Deshmukh et al., 2019). A summary is provided in the figure below:

Not shown in manuscript

Figure not shown in manuscript: the cartoon is adapted from Deshmukh et al. (see Ref.). On the left, numbers and % of proteins detected by Deshmukh et al. by LC-MS/MS on the conditioned media of primary adipocytes before and after filtering for secretion (classical – SignalP – and non-classical – SecretomeP). A very similar % of secreted proteins is identified in our total detected protein list when only filtering them according to SignalP and Matrisome. If SecretomeP-identified proteins were included, the % of secreted proteins would rise to 40%.

Figure adapted from Deshmukh et al. 2019, Cell Metab.

- 2) The quality of our secretome data is confirmed by:
 - i) enrichment of intracellular/secreted proteins in lysates/conditioned media respectively (see Fig. 1d);
 - ii) specific detection of soluble isoforms of cell-adhesion molecules in conditioned media (see Fig. 2f). An additional figure is shown here for the attention of the reviewer (please see **central panel in figure below**);
 - iii) their release is due to an active, regulated shedding mechanism from cell-membrane as supported by decreased detection of soluble fragments in conditioned media upon tissue explants incubation with MMP-inhibitor GM6001 (see Fig. 6f and **supplementary Fig. 4d**). An additional figure is shown here for the attention of the reviewer (please see **right panel in figure below**).

Figure 1d

Not shown in Manuscript

Not shown in Manuscript

Quality assessment of protocol for secretomes collection/analysis. To assess the quality of our protocol, we demonstrated that i) intracellular/secreted proteins are enriched in AT lysates/supernatants respectively (left, Fig. 1d in manuscript); ii) only the lightest, soluble isoforms of selected cell-adhesion molecules are detected by immunoblotting in AT conditioned media (C.M., center, not shown in manuscript, see also Fig. 2f in manuscript); iii) the detection of soluble cell-adhesion molecules in AT conditioned media is reduced/abolished upon incubation of tissue explants with the broad-spectrum MMP inhibitor GM6001 (right, figure not shown in manuscript).

- 3) Proteomics measured the abundances of lactate dehydrogenase (LDHA) and high mobility group box 1 (HMGB1), whose release in the conditioned media are well-established markers of cell death (Bell et al., 2006)(Cummings and Schnellmann, 2021). As shown below, LDHA and HMGB1 levels are comparable across samples, suggesting a low background release in all AT depots. In contrast, well-established secreted adipokines – adiponectin, PAI-1, visfatin, resistin, adipin and RBP-4 – clearly demonstrate that the different secretory activity of the various AT depots is maintained during 24h of *ex vivo* incubation.

Figure not shown in manuscript: relative abundances of established markers of cell death (HMGB1 and LDHA, left) and established secreted adipokines (adiponectin, visfatin, adipin, plasminogen activator inhibitor 1 – PAI-1 –, resistin and retinol binding protein 4 – RBP-4, right) detected by LC-MS/MS in AT explants conditioned media after 24 hours *ex vivo* incubation in DMEM-F12. One-way ANOVA followed by Bonferroni correction was performed.

As a related matter, for the comparisons between depots and models, it isn't clear how the authors are normalizing data. Do they analyze an equivalent mass of tissue? This should be discussed and justified in detail.

Additional details have now been added to the Methods section. Data normalisation was performed at every experimental step to account for any potential variation across samples:

- 1) Normalisation at volume/tissue level: PVAT depots are smaller than non-PVAT ones and the incubation volume was adjusted according to adipose tissue mass.
- 2) Normalisation at protein level: for proteomics, conditioned media were concentrated, and their protein content was quantified by BCA. The same amount of protein (10 µg) was used for LC-MS/MS analysis in line with previous proteomics analysis of the secretome of tissue explants (Alvarez-Llamas et al., 2007)(Hartwig et al., 2014)(Roca-Rivada et al., 2012)(de Wit et al., 2014)(de la Cuesta et al., 2012).
- 3) Normalisation at peptide level: after LC-MS/MS analysis, data were normalised by the total number of peptides identified in each sample.

2) Cellular origin of Negr1. While the authors have now provided adipose tissue fractionation data, this does not necessarily mean Negr1 is exclusively produced by adipocytes (or even produced by adipocytes at all). The authors should make this point more convincingly by either (a) differentiating SVF into adipocytes and showing that Negr1 mRNA is enriched in adipocytes and that Negr1 protein is secreted into the conditioned media by these adipocytes or (b) looking at Negr1 mRNA levels in published scRNA Seq and scNuc Seq datasets from adipose tissue. From the data provided, I am not at all convinced that Negr1 comes from adipocytes as opposed to from nerves innervating adipose tissue and associated vasculature.

We would argue that demonstrating Negr1 protein expression in primary adipocytes directly isolated from *ex vivo* fat explants represents more reliable evidence than differentiating adipocytes from precursors *in vitro*. Freshly isolated adipocytes reflect protein expression *in vivo*. Differentiating SVF to adipocytes could instead be biased by the differentiation process *in vitro*.

The protocol we have used for AT fractionation is a well-established method to separate primary adipocytes (floating) from SVF cells (pellet, see **picture below**) (Zhang et al., 2018)(Sassmann-Schweda et al., 2016)(Cutchins et al., 2012)(Fujiwara et al., 2012). We show the total absence of Negr1 protein in the SVF, while it is detected in the primary adipocytes fraction from all 3 depots analysed, together with the specific-adipocyte marker perilipin (see **Fig. 6d**, also shown **below**).

Furthermore, Negr1 expression has already been confirmed by others in isolated primary murine adipocytes (Joo et al., 2019) as well as in differentiated human adipocytes (Bernhard et al., 2013). Negr1 expression in adipocytes is also reported in the Human Protein Atlas among the 12,526 proteins detected by transcriptome analysis in human adipocytes.

Not shown in manuscript

Figure 6d

Figure 6e

Supplementary Figure 4c

Not shown in manuscript

Negr1 is enriched in adipocytes and is not expressed by sympathetic neurons. Left picture shows the 2 separated fractions obtained after digestion and fractionation of AT depots dissected from wt mice. In Fig. 6d, Negr1 protein is only detected in the floating adipocytes fraction (also positive for the adipocyte marker perilipin) but not in the stromal vascular fraction (SVF). Fig. 6e shows the total absence of Negr1 in primary murine sympathetic neurons lysates, which on the other hands do express cell adhesion molecules Ncam1 and Chl1 (Supp. Fig. 4c), in live with evidence obtained by staining PVAT sections (see Fig. 2d in manuscript). Bottom screenshot from Human Protein Atlas shows the detection of Negr1 in human adipocytes.

We can rule out that Negr1 is expressed by sympathetic nerves:

- 1) Negr1 is undetectable by immunoblotting of primary sympathetic neurons lysates, expressing the sympathetic marker tyrosine hydroxylase (TYR.H., see Fig. 6e, also shown above).
- 2) Primary sympathetic neurons on the other hand do express neuronal cell adhesion molecules Ncam1 and Chl1 (see Supplementary Fig. 4c, also shown above), in line with our evidence *in vivo* (see Fig. 2d).
- 3) We also provide below some of the z-stacks composing Fig. 7a and another section from wt TH PVAT, which better display the lack of a complete co-localization between Negr1 and TYR.H. Although the majority of Negr1 signal shows no overlap with the sympathetic marker, a clear accumulation in juxtaposition with neurites is observed in discrete areas (see white framed enlargements in Fig. 7a, below in the next page).

Tyrosine hydroxylase and Negr1 in TH PVAT sections from wt mice. The 2 panels show some of the confocal stacks composing Fig. 7a (left) and another section from wt TH PVAT (right). The lack of a complete correspondence between the tyrosine hydroxylase (TYR.H.) and Negr1 signal reinforces the notion that Negr1 is not expressed by sympathetic axons. This is in addition to Fig. 6e, showing a complete absence of Negr1 protein in primary sympathetic neurons lysates, and Fig. 6d, in which Negr1 is only detected in the fraction containing primary adipocytes after AT explants digestion and fractionation. Scale bars are 50 μ m.

Finally, I am quite concerned that the imaging for Negr1 in Figure 7a entirely represents background signal. It looks like the tissue is diffusely stained, and the authors need to do a control with tissue that doesn't express Negr1 or secondary antibody alone to make it convincing that this is real signal.

We have now added an **additional panel in Fig. 7a** showing the staining obtained after incubating the same sections with the secondary antibodies alone. Separate negative controls are provided for Negr1 (anti-GOAT) and tyrosine hydroxylase (TYR.H., anti-RB). Given that Negr1 is also released from membranes in a soluble form, it is expected that the staining is more diffused than the one for TYR.H..

Figure 7a. A panel showing a TH PVAT section only incubated with fluorescent 2ry antibodies has now been added to Fig. 7a. Scale bar: 50 μ m.

3) Negr1 function. The authors have presented new experiments towards a function for Negr1, but a number of important issues remain, and the conclusions are vastly overstated. First, in Figure 7e, they claim that adding Negr1 to CM from ob/ob explants prevents neurodegeneration. The claim here and elsewhere regarding neurodegeneration is entirely based on subtle and non-quantitative morphological alterations. Molecular markers of neurodegeneration need to be analyzed.

The appearance of swellings/blebbings along neurites is a well-established marker of neurodegeneration both *in vitro* and *in vivo*, and has been employed as a quantitative degeneration index in several high-impact papers, many of them in journals specific to the neuroscience field (Singh et al., 2008, *Nat. Neuroscience*)(Park et al., 2010, *Nat. Neuroscience*)(Yong et al., 2020, *Sci. Rep.*)(Nikolaev et al., 2009, *Nature*)(Wang et al., 2005). Moreover, quantification of swellings was performed in several fields from multiple biological replicates in a blind fashion, and this further reinforces the reliability of our data. Molecular markers and pathways vary with regards to the specific cell death mechanism involved in the neurodegeneration, which goes beyond the scope of this work.

In response to the reviewer's criticism, lower magnification images of β 3-tubulin staining have now been added to the supplementary section (**New Supplementary Fig. 4b and 5b**). It is clearly apparent that the axonal status can be delineated in different experimental conditions.

New Supplementary Fig. 4b

New Supplementary Fig. 5b

New Supplementary Figures 4b and 5b. Representative 10X pictures of microfluidics axonal chambers incubated for 24 h with the indicated conditioned media (C.M.) +/- 500 ng/ml recombinant Negr1, fixed and stained for β3-tubulin. Scale bars: 100 μm.

Second, if Negr1 is a neurotrophic factor, how do the authors explain their data in Figure S4D showing highest levels in depots that are least innervated?

The focus of the present paper is the comparison between wt and *ob/ob* mice. Variation in physiological innervation across different AT depots must be the subject of future studies. Addressing differences in environment for axonal growth/maintenance by AT depot-specific combinations of growth factors and conditions would be beyond the scope of the present manuscript. However, the lack of correspondence between Negr1 abundance and innervation extent further corroborates our point that Negr1 in AT does not originate from sympathetic nerves.

Third, the new in vivo experiment in Figure 7f-i needs much more corroboration. Specifically, what levels of Negr1 were achieved and was Negr1 only elevated locally or also systemically?

We have addressed this comment by including a new Supplementary Fig. 5c (see below). We show by immunoblotting on SC AT lysates that the local levels of Negr1 are significantly increased by an average of 3-fold in mice receiving the recombinant protein, even after extensive rinsing of the tissue explants before lysis. Conversely, no significant difference is detected for Negr1 abundances in other AT depots from the same mice (i.e., VI AT and BAT, see new Supplementary Fig. 5c), nor in the liver (shown below, not included in manuscript).

New Supplementary Fig. 5c

Not shown in manuscript

Negr1 abundances in organs from minipumps-implanted mice.

AT samples from mice receiving either recombinant Negr1 or saline for 2 weeks were harvested, washed, lysed, quantified and immunostained for Negr1 (Supplementary Fig. 5c, top). While a significant increase of Negr1 was detected locally in SC AT samples of mice receiving Negr1, this was not the case for other distal AT depots (BAT and VI WAT), nor for the liver (left, not shown in manuscript).

Proper delivery of Negr1 was also confirmed by i) checking the function of catheter-connected minipumps during the priming phase before implantation and ii) by measuring the remaining volume within minipumps after removal from animals as recommended by the manufacturer. This is now stated in the methods section.

Also, the TH imaging needs to be supported by tissue imaging for nerve fibers, as the authors have done elsewhere in the manuscript.

For technical reasons, we cannot include imaging of nerve fibers in *ob/ob* SC AT. Nerve fibers in *ob/ob* SC AT are so dispersed that even with a 5X objective we cannot reliably identify and quantify them (see figure below). Importantly, *ob/ob* SC AT is too fatty for cryosectioning and can only be processed for histology when embedded in paraffin. However, the antibody used for TYR.H. imaging works very well for cryo- but not for paraffin sections – see paraffin BAT sections used as positive control below and cryosections employed throughout the manuscript for comparison.

Not shown in manuscript

Sympathetic fibers staining in AT paraffin sections. Tyrosine hydroxylase (TYR.H., red) was stained together with Perilipin (green) in 8 μ m-thick paraffin sections following antigen retrieval. BAT from wt mice was used as positive control, due to its dense innervation. Despite some TYR.H. signal was visible in BAT sections, the antibody efficiency is much lower than in cryosections (employed throughout the manuscript). Moreover, the extreme adipocyte hypertrophy in SC AT from *ob/ob* mice results in an even lower density of neurites, which can not be reliably identified and quantified even with a very low magnification objective. Scale bar: 100 μ m.

Instead, we have now included a **new Fig. 7i** (shown below). We stained the same lysates shown in Fig. 7i for the axonal marker β 3-tubulin (as previously done for human epicardial fat samples in Fig. 4g). β 3-tubulin is also significantly increased in the SC AT of mice administered with Negr1 corroborating our results for TYR.H..

New Figure 7i

New Figure 7i. The sympathetic marker tyrosine hydroxylase (TYR.H.) and the pan-axonal marker β 3-tubulin are significantly increased in the SC AT lysates from *ob/ob* mice administered with recombinant Negr1 compared to mice only receiving saline.

The authors overall make excessive claims based on TH westerns. This is a concern because the TH antibody used here has been shown to detect similar sized bands in both control and TH KO samples and thus may not be reliable (Fischer et al, Nat Med 2017, Fig S1).

We respectfully disagree. We are convinced that this is not due the quality of the primary antibody: see **middle and right panel** of our Western blots compared to Nat Med Fig. S1 in left panel below.

AB152 tyrosine hydroxylase antibody specificity. While several unspecific bands can be observed in the Supp. Fig. 1 from Fischer et al. mentioned by the reviewer (left), in our hands this antibody only produces a very specific band of the proper molecular weight (center and right: whole membranes shown in Fig. 4a and Supp. Fig. 3d in our manuscript).

- 1) The reviewer is right that the Fig. S1 from the *Nat Med* paper (Fischer et al., 2017) is indeed concerning (reproduced as **left panel** in figure **above**). It shows numerous (potentially unspecific) bands and does not include positive controls. However, these bands are likely to originate from unspecific binding of the secondary antibody to endogenous murine heavy immunoglobulins. Given the characteristic Mw pattern with signals at 50 kDa, dimers just above 100 kDa and multimers at 250 kDa, negative controls with secondary antibodies alone should have been performed. Also, the “TH KO” mouse used in (Fischer et al., 2017) is not a constitutive KO for tyrosine hydroxylase, so a residual presence of tyrosine hydroxylase cannot be ruled out.
- 2) In our hands, the TYR.H. antibody (AB152 from Millipore) has excellent specificity for immunoblotting. In our Western blots this antibody only produces a single, distinct band of the right molecular weight. Brain lysates were always loaded on the side of samples as positive control.
- 3) The same antibody AB152 was also used for immunoblotting as a specific sympathetic marker in several high-impact papers (Chi et al., 2018, *Cell Metab.*)(Peyrou et al., 2020, *Nat. Commun.*)(Wang et al., 2020, *Nature*)(Kimura et al., 2007, *Circ. Res.*).

A list of all relevant references is given below.

REFERENCES

- Ali Khan, A., Hansson, J., Weber, P., Foehr, S., Krijgsveld, J., Herzig, S., and Scheideler, M. (2018). Comparative secretome analyses of primary murine white and brown adipocytes reveal novel adipokines. *Mol. Cell. Proteomics* 17, 2358–2370.
- Alvarez-Llamas, G., Szalowska, E., de Vries, M.P., Weening, D., Landman, K., Hoek, A., Wolffenbuttel, B.H.R., Roelofsen, H., and Vonk, R.J. (2007). Characterization of the human visceral adipose tissue secretome. *Mol. Cell. Proteomics* 6, 589–600.
- Bell, C.W., Jiang, W., Reich, C.F., and Pisetsky, D.S. (2006). The extracellular release of HMGB1 during apoptotic cell death. *Am. J. Physiol. - Cell Physiol.* 291, 1318–1325.
- Bernhard, F., Landgraf, K., Klötting, N., Berthold, A., Büttner, P., Friebe, D., Kiess, W., Kovacs, P., Blüher, M., and Körner, A. (2013). Functional relevance of genes implicated by obesity genome-wide association study signals for human adipocyte biology. *Diabetologia* 56, 311–322.
- Chi, J., Wu, Z., Choi, C.H.J., Nguyen, L., Tegegne, S., Ackerman, S.E., Crane, A., Marchildon, F., Tessier-Lavigne, M., and Cohen, P. (2018). Three-Dimensional Adipose Tissue Imaging Reveals Regional Variation in Beige Fat Biogenesis and PRDM16-Dependent Sympathetic Neurite Density. *Cell Metab.* 27, 226-236.e3.
- Cummings, B.S., and Schnellmann, R.G. (2021). Measurement of Cell Death in Mammalian Cells. *Curr. Protoc.* 1, 1–30.
- Cutchins, A., Harmon, D.B., Kirby, J.L., Doran, A.C., Oldham, S.N., Skaffen, M., Klibanov, A.L., Meller, N., Keller, S.R., Garmey, J., et al. (2012). Inhibitor of differentiation-3 mediates high fat diet-induced visceral fat expansion. *Arterioscler. Thromb. Vasc. Biol.* 32, 317–324.
- Deshmukh, A.S., Peijs, L., Beaudry, J.L., Jespersen, N.Z., Nielsen, C.H., Ma, T., Brunner, A.D., Larsen, T.J., Bayarri-Olmos, R., Prabhakar, B.S., et al. (2019). Proteomics-Based Comparative Mapping of the Secretomes of Human Brown and White Adipocytes Reveals EPDR1 as a Novel Adipokine. *Cell Metab.* 30, 963-975.e7.
- Fischer, K., Ruiz, H.H., Jhun, K., Finan, B., Oberlin, D.J., Van Der Heide, V., Kalinovich, A. V., Petrovic, N., Wolf, Y., Clemmensen, C., et al. (2017). Alternatively activated macrophages do not synthesize catecholamines or contribute to adipose tissue adaptive thermogenesis. *Nat. Med.* 23, 623–630.
- Fujiwara, K., Hasegawa, K., Ohkumo, T., Miyoshi, H., Tseng, Y.H., and Yoshikawa, K. (2012). Necdin controls proliferation of white adipocyte progenitor cells. *PLoS One* 7.

- Hartwig, S., Goeddeke, S., Poschmann, G., Dicken, H.D., Jacob, S., Nitzgen, U., Passlack, W., Stühler, K., Ouwens, D.M., Al-Hasani, H., et al. (2014). Identification of novel adipokines differentially regulated in C57BL/Ks and C57BL/6. *Arch. Physiol. Biochem.* *120*, 208–215.
- Joo, Y., Kim, H., Lee, S., and Lee, S. (2019). Neuronal growth regulator 1-deficient mice show increased adiposity and decreased muscle mass. *Int. J. Obes.* *43*, 1769–1782.
- Kimura, K., Ieda, M., Kanazawa, H., Yagi, T., Tsunoda, M., Ninomiya, S.I., Kurosawa, H., Yoshimi, K., Mochizuki, H., Yamazaki, K., et al. (2007). Cardiac sympathetic rejuvenation: A link between nerve function and cardiac hypertrophy. *Circ. Res.* *100*, 1755–1764.
- de la Cuesta, F., Barderas, M.G., Calvo, E., Zubiri, I., Maroto, A.S., Darde, V.M., Martín-Rojas, T., Gil-Dones, F., Posada-Ayala, M., Tejerina, T., et al. (2012). Secretome analysis of atherosclerotic and non-atherosclerotic arteries reveals dynamic extracellular remodeling during pathogenesis. *J. Proteomics* *75*, 2960–2971.
- Nikolaev, A., McLaughlin, T., O’Leary, D.D.M., and Tessier-Lavigne, M. (2009). APP binds DR6 to trigger axon pruning and neuron death via distinct caspases. *Nature* *457*, 981–989.
- Park, K.J., Grosso, C.A., Aubert, I., Kaplan, D.R., and Miller, F.D. (2010). P75NTR-dependent, myelin-mediated axonal degeneration regulates neural connectivity in the adult brain. *Nat. Neurosci.* *13*, 559–566.
- Peyrou, M., Cereijo, R., Quesada-López, T., Campderrós, L., Gavaldà-Navarro, A., Liñares-Pose, L., Kaschina, E., Unger, T., López, M., Giral, M., et al. (2020). The kallikrein–kinin pathway as a mechanism for auto-control of brown adipose tissue activity. *Nat. Commun.* *11*, 1–16.
- Roca-Rivada, A., Al-Massadi, O., Castelao, C., Senín, L.L., Alonso, J., Seoane, L.M., García-Caballero, T., Casanueva, F.F., and Pardo, M. (2012). Muscle tissue as an endocrine organ: Comparative secretome profiling of slow-oxidative and fast-glycolytic rat muscle explants and its variation with exercise. *J. Proteomics* *75*, 5414–5425.
- Sassmann-Schweda, A., Singh, P., Tang, C., Wietelmann, A., Wettschreck, N., and Offermanns, S. (2016). Increased apoptosis and browning of TAK1-deficient adipocytes protects against obesity. *JCI Insight* *1*.
- Singh, K.K., Park, K.J., Hong, E.J., Kramer, B.M., Greenberg, M.E., Kaplan, D.R., and Miller, F.D. (2008). Developmental axon pruning mediated by BDNF-p75NTR-dependent axon degeneration. *Nat. Neurosci.* *11*, 649–658.
- Wang, J., Zhai, Q., Chen, Y., Lin, E., Gu, W., McBurney, M.W., and He, Z. (2005). A local mechanism mediates NAD-dependent protection of axon degeneration. *J. Cell Biol.* *170*, 349–355.
- Wang, P., Loh, K.H., Wu, M., Morgan, D.A., Schneeberger, M., Yu, X., Chi, J., Kosse, C., Kim, D., Rahmouni, K., et al. (2020). A leptin–BDNF pathway regulating sympathetic innervation of adipose tissue. *Nature* *583*, 1–6.
- de Wit, M., Kant, H., Piersma, S.R., Pham, T. V., Mongera, S., van Berkel, M.P.A., Boven, E., Pontén, F., Meijer, G.A., Jimenez, C.R., et al. (2014). Colorectal cancer candidate biomarkers identified by tissue secretome proteome profiling. *J. Proteomics* *99*, 26–39.
- Yong, Y., Gamage, K., Cushman, C., Spano, A., and Deppmann, C. (2020). Regulation of degenerative spheroids after injury. *Sci. Rep.* *10*, 1–17.
- Zhang, R., Gao, Y., Zhao, X., Gao, M., Wu, Y., Han, Y., Qiao, Y., Luo, Z., Yang, L., Chen, J., et al. (2018). FSP1-positive fibroblasts are adipogenic niche and regulate adipose homeostasis. *PLoS Biol.* *16*, 1–21.

REVIEWER COMMENTS

Reviewer #1 (Remarks to the Author):

In my most recent comments, I raised concerns about the quality of the secretome data, the cellular origin of Negr1, and Negr1 function. While I appreciate that the authors have attempted to address each of these issues, my concerns remain.

1) Secretome data

a) In the rebuttal, the authors present a figure (bottom right of page 1) arguing that treatment with an MMP inhibitor blocks the detection of soluble cell adhesion molecules in the CM. Based on the Ponceau stain, the wells treated with GM6001 are underloaded, so this is not convincing.

b) On the top of page 2 of the rebuttal, the authors show relative levels of HMGB1 and LDHA. Because they are comparable, they argue that this suggests a “low background release in all AT depots.”

However, because these are relative levels, that point cannot be made. It is possible that there is actually a comparable and high level of cell death across samples.

2) Cellular origin of Negr1

a) The sections shown on page 4 of the rebuttal now stained with anti-GOAT only show essentially the same staining pattern as the experimental samples with slightly diminished intensity. To me, this makes it more likely that the staining is in fact background.

3) Negr1 function

a) The experiment shown at the bottom of Figure 5 of the rebuttal does not make sense. If the authors are delivering recombinant Negr1 through an osmotic pump and then extensively rinsing tissues extensively before making lysates and doing blots, then the Negr1 detected is most likely coming from endogenous tissue production and not from the exogenous source.

b) The authors are unable to detect nerve fibers in ob/ob SQ fat. Several other publications have managed to do so.

c) With regard to the TH antibody, the figure on page 7 of the rebuttal shows less background, but there is no negative control. As a result, it is actually possible that the bands detected are false positives.

Reviewer #1 (Remarks to the Author):

In my most recent comments, I raised concerns about the quality of the secretome data, the cellular origin of Negr1, and Negr1 function. While I appreciate that the authors have attempted to address each of these issues, my concerns remain.

1) Secretome data.

a) In the rebuttal, the authors present a figure (bottom right of page 1) arguing that treatment with an MMP inhibitor blocks the detection of soluble cell adhesion molecules in the CM. Based on the Ponceau stain, the wells treated with GM6001 are underloaded, so this is not convincing.

We would like to stress that in all our conditioned media protein concentrations were not only quantified by a colorimetric assay but also at the peptide level by using the total ion currents from our mass spectrometry analysis. Thus, the total protein amounts are very accurate. The same protein amount was loaded in all wells. To address the reviewer's concern of "underloading", a larger Ponceau crop has now been added to the figure below.

It is obvious from these blots that:

- i) There are very pronounced differences for the soluble cell adhesion molecules (Negr1, L1cam and Ncam1) upon GM6001 treatment.
- ii) In contrast, the levels of PAI1 and adiponectin are not affected by GM6001 as their release is not mediated by MMP cleavage.

Thus, the differences in soluble cell adhesion molecules cannot be explained by loading differences. Moreover, the results for Negr1 were reproduced in conditioned media of TH PVAT, AR PVAT (see Fig. 6f) and SC AT (see Supplementary Fig. 4e).

b) On the top of page 2 of the rebuttal, the authors show relative levels of HMGB1 and LDHA. Because they are comparable, they argue that this suggests a "low background release in all AT depots." However, because these are relative levels, that point cannot be made. It is possible that there is actually a comparable and high level of cell death across samples.

The reviewer is referring to a figure presented on page 2 of our previous rebuttal. In this figure, we have addressed the reviewer's previous concern whether tissue viability is similar across samples and whether we observe variation in intracellular protein release by cell

death. In response to the reviewer's latest request, we have now measured LDH activity in BAT and VI AT samples at the beginning and at the end of the 24h incubation period (figure below). The LDH activity in conditioned media does not significantly increase in AT explants during the 24h incubation. Addition of 1% Triton served as a positive control (N=3 biological replicates per group, paired t-test). These results clearly demonstrate that the tissue viability is preserved in AT explants during the 24h incubation in serum-free medium for secretomes analysis.

BAT and VI AT samples (N=3 biological replicates) were harvested from wt male mice and processed for conditioned media collection as described in Methods section. In a subset of samples, 1% final Triton was added to the medium after 1 hour for the remaining 23 hours as a positive control for cytotoxicity. LDH activity was measured in the conditioned media at 1h (before adding Triton) and at 24 hours.

2) Cellular origin of Negr1.

- a) The sections shown on page 4 of the rebuttal now stained with anti-GOAT only show essentially the same staining pattern as the experimental samples with slightly diminished intensity. To me, this makes it more likely that the staining is in fact background.

We disagree with the reviewer and include the figure of our previous rebuttal below (Figure 7a). As mentioned before, Negr1 is present both as a membrane-anchored and as a cleaved, soluble form. Thus, it is expected that a Negr1 signal is detected not only in discrete regions on the cell membrane but also more diffusely in the extracellular space. The discrete regions were highlighted in our images with white frames. No such discrete staining patterns on the cell surface was observed with anti-Goat secondary antibody only. Similarly, the signal in the extracellular space was a much stronger with the Negr1 primary antibody than the secondary anti-Goat antibody only – see merged images. All microscope and ImageJ parameters were kept consistent across all samples for acquisition as well as for the analysis of the displayed images.

Figure 7a. A panel showing a TH PVAT section only incubated with fluorescent 2ry antibodies has now been added to Fig. 7a. Scale bar: 50 μ m.

3) Negr1 function.

a) The experiment shown at the bottom of Figure 5 of the rebuttal does not make sense. If the authors are delivering recombinant Negr1 through an osmotic pump and then extensively rinsing tissues extensively before making lysates and doing blots, then the Negr1 detected is most likely coming from endogenous tissue production and not from the exogenous source.

An additional figure (new Supplementary Figure 5d, also shown below, left panel) has now been added to the manuscript showing that the relative mRNA levels of Negr1 are unchanged in the SC WAT of mice receiving recombinant Negr1 via minipump, excluding a contribution of endogenous Negr1 to the differences in proteins levels observed in the same tissues (Supplementary Fig. 5c). This indicates that exogenous Negr1 is retained within the tissue. This retention is also observed *in vitro*, where Negr1 is detected in the lysates of sympathetic neurons pre-incubated with recombinant Negr1 (1 ng/ml for 24 hours) and washed 3 times with sterile PBS before lysis. In contrast, Negr1 is completely undetectable in untreated samples, thus ruling out neurons as a major endogenous source (see below, central panel). Similarly, no differences in the level of Negr1 are observed in other tissues from mice receiving recombinant Negr1 locally in the SC WAT (i.e., BAT, VI AT and liver, see Supplementary Fig. 5c and figure at page 5 of previous rebuttal). Negr1 is expected to exert its neurotrophic effect by binding to sympathetic neurons and leading to the activation of the MAPK/ERK pathway as shown below (right panel). Thus, the significant difference observed in the SC AT after minipump infusion of Negr1 can be attributed to the exogenous Negr1 which remains in the tissue and is not removed by tissue rinsing before processing.

New Supplementary Fig. 5d

Left: Relative mRNA levels of *Negr1* normalized on *Tbp* in SC WAT lysates of mice receiving for 2 weeks either recombinant Negr1 or saline at the level of SC WAT via catheter-connected minipumps. Data are shown as mean \pm SEM, N=5 biological replicates per group, unpaired, two-tailed t-test. **Centre:** Primary sympathetic neurons were incubated in absence/presence of 1 ng/ml recombinant Negr1 for 1 hour. At the end of the incubation, each well was washed 3 times with sterile pre-warmed PBS, then lysed, quantified and loaded for immunoblotting. This Western blot clearly shows that despite extensive washes, recombinant Negr1 still remains clearly detectable in treated (but not untreated) samples, suggesting that some Negr1 remains bound to sympathetic neurons. **Rigth:** Representative Western blot and quantification of 3 independent experiments showing that addition of recombinant Negr1 induces the phosphorylation of ERK in primary murine sympathetic neurons. In more detail, neurons were deprived from NGF for 7 hours and then either NGF, Negr1 or a combination of the 2 factors were added to the cells for 10 minutes. Neurons were then lysed and processed by Western blot.

b) The authors are unable to detect nerve fibers in ob/ob SQ fat. Several other publications have managed to do so.

The point we tried to make was that the SC AT innervation in *ob/ob* mice is so dispersed that imaging would not be as reliable for accurate quantification compared to immunoblotting of SC AT tissue extracts. The immunoblot data provided in Fig. 7i

provide the quantification of two distinct axonal markers and ample evidence for a quantitative difference between the 2 groups. Moreover, we demonstrate a good correspondence between tyrosine hydroxylase intensity in immunoblot and sympathetic innervation density in frozen adipose sections as shown in Fig. 2g and 2h. Finally, the quality of AB152 for immunoblotting has been further validated as shown in the rebuttal to the next point raised by the reviewer.

c) With regard to the TH antibody, the figure on page 7 of the rebuttal shows less background, but there is no negative control. As a result, it is actually possible that the bands detected are false positives.

We think the quality of our Western blots is outstanding. Below we include 2 representative Western blots stained for tyrosine hydroxylase (AB152) with the positive and negative controls as recommended by the manufacturer:

[https://www.merckmillipore.com/GB/en/product/Anti-Tyrosine-Hydroxylase-Antibody,MM NF-AB152](https://www.merckmillipore.com/GB/en/product/Anti-Tyrosine-Hydroxylase-Antibody,MM%20NF-AB152)

i.e., brain (as positive control) and liver (as negative control). Tyrosine hydroxylase is clearly detected in brain samples but not in liver samples. In addition, we have used spleen extracts as negative control to further rule out false positive staining.

Western blots showing tyrosine hydroxylase staining obtained in distinct murine and human samples. While a clear, single band is detected in murine brain, BAT, AR PVAT and in human epicardial fat (EAT), tyrosine hydroxylase is not detected in murine spleen and liver (also reported to work as negative control in the datasheet of AB152 antibody).

REVIEWERS' COMMENTS

Reviewer #1 (Remarks to the Author):

The rebuttal and revised manuscript has now satisfactorily addressed the concerns raised in my last review.

We would like to thank Reviewer #1 for acknowledging that all his concerns have been addressed in our last version of the manuscript and in the rebuttal.